

# Multi-decadal past winter temperature, precipitation and snow cover information over the European Alps using multiple datasets

Diego Monteiro[1] and Samuel Morin[1,2]

[1]Univ. Grenoble Alpes, Université de Toulouse, Météo-France, CNRS, CNRM, Centre d'Etudes de la Neige, 38000 Grenoble, France
[2]CNRM, Météo-France, CNRS, Université de Toulouse, Toulouse, France

**Correspondence:** Diego Monteiro (diego.monteiro@meteo.fr)

**Abstract.**

Assessing past distributions, variability and trends of the mountain snow cover and its first order drivers, temperature and precipitation, is key for a wide range of studies and applications. In this study, we compare the results of various modelling systems (global and regional reanalyses ERA5, ERA5-Land, ERA5-Crocus, CERRA-Land, UERRA MESCAN-SURFEX, MTMSI, and regional climate model simulations CNRM-ALADIN and CNRM-AROME driven by the global reanalysis ERA-Interim) against observational references (in-situ, kriged datasets and satellite observations) across the European Alps, from 1950 to 2020. The comparisons are performed in terms of monthly and seasonal snow cover variables (snow depth and snow cover duration) and their main atmospherical drivers (near-surface temperature and precipitation). We assess multi-annual averages of regional and sub-regional mean values, their inter-annual variations, and trends over various time scales.

CNRM-AROME and CNRM-ALADIN simulations, and ERA5-Land exhibit an overestimation of the snow accumulation during winter, increasing with elevations. ERA5, ERA5-Crocus, MESCAN-SURFEX, CERRA-Land and MTMSI offer a satisfying description of the monthly snow evolution albeit a spatial comparison against satellite observation indicates that all datasets overestimate the snow cover duration of the snow cover, especially the melt-out date.

The analysis of the inter-annual variability and trends indicate that modelling snow cover dynamics remain complex across multiple scales, that none of the models evaluated here fully succeed to reproduce, compared to observational reference datasets. Indeed, while most of the evaluated model outputs perform well at representing the inter-annual to multi-decadal winter temperature and precipitation variability, they often fail to address the variability of the snow depth and snow cover duration. We discuss several artifacts potentially responsible for incorrect long-term climate trends in several reanalysis products (ERA5 and MESCAN-SURFEX), which we attribute primarily to the heterogeneities of the observation datasets assimilated.

Reference datasets and some of the evaluated datasets provides past trends in line with current available literature. Over the last 50 years (1968-2017) at a regional scale, the European Alps have experienced a winter warming of 0.3 to 0.4°C per decade, a weak reduction of winter precipitation, and a substantial decrease of the snowpack characteristics, with a decline of the winter snow depth and the snow cover duration reaching -10% per decade and -10 days per decade, respectively, especially at low and intermediate elevations.





Overall, we show that no modelling strategy outperforms all others within our sample, and that upstream choices (horizontal resolution, heterogeneity of the observations used for data assimilation in reanalyses, coupling between surface and atmosphere, level of complexity and configuration of the snow scheme etc.) have great consequences on the quality of the datasets and their potential use. Despite their limitations, in many cases these modeling outputs can be used to characterize the main features of the mountain snow cover for a range of applications.

## 1    Introduction

Emissions of greenhouse gases by industrial societies since the 1850s have led to an increase in the global mean surface temperature of 1.07°C (0.8°C-1.3°C), 1.59°C (1.34-1.83 °C) over land (Zhongming et al., 2021) inducing a series of modifications in the components of the Earth system. In mountainous areas, composed of a large number of systems and environments sensitive to climate change, the temperature rise has already led to major impacts (Hock et al., 2019). Observations from the past

decades generally show a decrease in glacier mass, a temperature rise of the permafrost, and a general decline of the snow cover duration on average by 5 days per decade at low elevation. These changes already impact water resources and agriculture in snow-dominated and glacier-fed river basins, as well as altering the magnitude and location of natural hazards in mountainous regions (Hock et al., 2019). While a wide range of physical changes in mountain regions are already well understood in general terms (Hock et al., 2019), some changes remain imperfectly characterized (Pepin et al., 2015, 2022), especially at the local

scale, as well as numerous impacts on a large variety of related domains (e.g. hydrology, ecology, natural hazards...).

Reliable observational data on essential climate variables (e.g. near surface temperature, precipitation, snow cover area, snow water storage... ) are critically needed to further investigate past changes, improve process understanding, and characterize the present state of the different systems and environments under a changing climate. Yet due to constraints (e.g. accessibility, extreme climate conditions...) on the installation and maintenance of a dense observational network, the historical and current

in-situ coverage is sparse over mountainous regions, specifically at high elevation, even in the European Alps, one of the most extensively studied mountain range in the world. Multiple approaches have been developed to complement the information from sparse in-situ observation networks and to gather informations about the past state of the climate system in mountain regions.

Remote sensing data have the advantage of almost exhaustive spatial coverage at a high resolution (down to a few tens of

meters of horizontal resolution). However, only a limited number of climate variables can be derived from them, with a short and generally partial temporal coverage that does not allow the reconstruction of past changes over the last century. MODIS (Justice et al., 1998), for example, provides records from February 2000 onwards, less than the conventional 30 years required to define a climatology, let alone a climate trend. Additionnally, the quality of remote sensing data is weakest in mountainous regions due to the complex topography compared to flatland (Largeron et al., 2020).

Reanalyses are generated using a numerical Weather prediction (NWP) model simulating the state of atmospheric and surface variables, using observational constraints through data assimilation. The aim of reanalyses is to provide information about the state of the atmosphere and its interfaces consistent with the observed chronology of meteorological events. Over the last





decade, a new series of global and regional reanalyses have been released, taking advantage of the rise of model performance and assimilation procedures, providing high resolution climate informations. Among them, ERA5 (Hersbach et al., 2020) and ERA5-Land (Muñoz-Sabater et al., 2021) are global reanalyses recently provided by the ECMWF (European Centre for Medium-Range Weather Forecasts), and already extensively used in a wide range of applications. UERRA-MESCAN-SURFEX and CERRA-Land are high resolution regional reanalyses resulting from a series of european project (EURO4M, UERRA, now implemented as part of the Copernicus Climate Change Service and named CERRA), taking advantage of their high resolution, and the use of a new surface analysis system MESCAN (Soci et al., 2016) to provide a robust estimation of surface variables over Europe. These new reanalyses are promising tools, but are still limited for some applications. Besides their high computational cost, they remain dependent on model limitations, and an assimilation system that can lead to spurious trends due in particular to the spatio-temporal heterogeneity of assimilated observations (Thorne and Vose, 2010; Vidal et al., 2010; Vernay et al., 2022).

Regional climate simulations forced by a larger scale reanalysis are continuous long-term simulations over a limited area. They are, by design, constrained to follow the large scale chronology of meteorological episodes, and avoid some of the issues induced by the assimilation steps of regional reanalyses, but inherit biases from the atmospherical model. Regional climate simulations driven by larger scale reanalyses are mostly used as a benchmarking tool to assess the reliability of climate simulations, used for climate projections. In Europe, climate simulations produced within the EURO-CORDEX framework have been used in various applications ranging from physical changes to climate change impacts (Jacob et al., 2014; Beniston et al., 2018; Terzago et al., 2017; Kotlarski et al., 2023). More recently, the EURO-CORDEX Flagship Pilot Study "Convection" has lead to the production of high resolution regional climate simulations using climate models at kilometer scale over a domain that covers the Alpine ridge (Coppola et al., 2020; Pichelli et al., 2021). These simulations have demonstrated their potential for the study of rare events such as high precipitation event (HPE) (Caillaud et al., 2021), as well as improving the representation of mountain variables such as temperature, precipitation, and snow cover over the Alps (Lüthi et al., 2019; Monteiro et al., 2022). A recent review from Lundquist et al. (2019) suggests that high resolution climate simulation are now able to produce a better estimate of meteorological variables over mountainous areas than gridded datasets based on in-situ observations, limited by the scarcity of in-situ observations and some of their observation limitations, e.g. snow precipitation wind-induced undercatch.

Several studies have focused on evaluating datasets and reanalyses in various contexts with the aim of outlining their adequate areas of use. Muñoz-Sabater et al. (2021) provide an extensive comparison of ERA5 and ERA5-Land against in-situ observations for a set of surface variables (near-surface air temperature, soil moisture, snow depth). Their study highlights that besides a clear added-value of ERA5-Land against ERA5 in the Western US considering the representation of snow, their comparison over Scandinavia and the Northern part of the Alps lead to more nuanced results that they attributed to the quality of ERA5, due to the density of the assimilated observation network in ERA5 in these regions. Isotta et al. (2015) and Bandhauer et al. (2022) focus on precipitation characteristics over the Alpine region from numerous datasets (ERA5, MESCAN, EURO4M-APGD, E-OBS... ). They report on a widespread overestimation of accumulated precipitation and wet-day frequency of ERA5 and UERRA against gridded observational datasets, and show that their local scale performances depends on the density of the rain-gauge network. Li et al. (2022) provide an intercomparison of snow depth from ERA5, ERA5-Land



and WRF climate simulations against remote sensing and observational datasets over the Tianshan Mountains in China and find that constrasting results. contrasting results. ERA5-Land is prone to lower errors (RMSE and ME) compared to ERA5 at low and intermediate elevation, but shows larger biases at high elevation. They both perform poorly regarding the annual evolution of snow, with an overestimated accumulation phase for ERA5 and an underestimation of the melting rate for ERA5-Land. Overall, the WRF climate simulation performs well at all elevations, and gives the closer estimates of the annual cycle of snow cover. Orsolini et al. (2019) study the ERA5 abilities to represents snow characteristics (snow depth, snow cover, snow duration... ) over the Tibetan Plateau (TP) and find that ERA5 strongly overestimate the amount and duration of snow cover over the TP, that they relate to the lack of assimilated observations in this region, as well as a strong overestimation of precipitations. Scherrer (2020) compares near-surface temperature inter-annual variability and trends for a set of reanalyses and gridded datasets (i.e. ERA5, MESCAN, E-OBS and COSMO-REA) against a gridded datasets for Switzerland and find that they all perform well at low elevations but have increasing errors in terms of trend and internal variability at high elevation. Kaiser-Weiss et al. (2019) perform a broad evaluation over Europe of multiple reanalyses (those from the UERRA project and COSMO-REA) for wind, solar radiation, precipitation and temperature and find that. The authors insisted on the predominant factors determining the quality of the dataset that is the number of assimilated observations inside a given area.

The above studies lead to nuanced results concerning the ability of recent reanalyses, gridded observation-based datasets and climate simulations to provide reliable long-term informations relevant to characterize mountain meteorological conditions and the snow cover state. None of them outperforms other datasets in every regions and analysed aspects of the climatology (i.e. mean values, seasonal patterns, spatial patterns, interannual variability, trend), but they hold promising potentials to complement in-situ observation records. Multiple factors are involved and strongly depend on the study area such as the quality of atmospheric forcing driving the land surface model, the number and quality of the assimilated observations and the inherited biases from the underlying model used (atmospherical model and land surface model). Thus, it is clear that extensive studies are needed to qualify the robustness of these emerging tools for appropriate use in a wide range of downstream scientific applications.

The objective of the present study is to compare the performance of different datasets from different modelling strategies in the European Alps, in order to provide the best possible estimate of the state of the snow cover, and its first order drivers, wintertime near surface temperature and precipitation. We investigate and evaluate mean seasonal and annual values, spatial variability and patterns, and interannual variability and trends over the last decades. We take advantage of the recent study by Matiu et al. (2021) providing a consolidated dataset of in-situ snow depth observations in the European Alps. We also exploit two gridded observational reference datasets LAPrec (Frei and Isotta, 2019) for the precipitation (specifically covering the Alpine region) and E-OBS for the near surface temperature (Cornes et al., 2018). By doing so, we aim to provide informations on the reliability of several commonly used and most recent reanalyses as well as other modelling approaches in the European Alps and provide estimates of climate trends of the variables.



## 2 Data and Methods

### 2.1 Study domain

Our study domain is the European Alps (see Figure 1a), using the boundaries of the Alpine Convention (Convention, 2020). We carry out analyses over the whole region, or for the four subregions following the HISTALP division from Auer et al. (2007). These four subregions correspond to four climatically homogeneous areas: the western side (northwest and southwest) influenced by the Atlantic and an Eurasian continental regime on the Eastern side. North-south border distinguishes the warmer and dryier mediterranean side on the south (southwest and southeast) and the wetter northern part (northwest and northeast), blocking most of the western lows. This division into four main subregions based on temperature, precipitation, air pressure, sunshine and cloudiness was recently confirmed to be relevant by Matiu et al. (2021) based on snow depth in-situ observations.

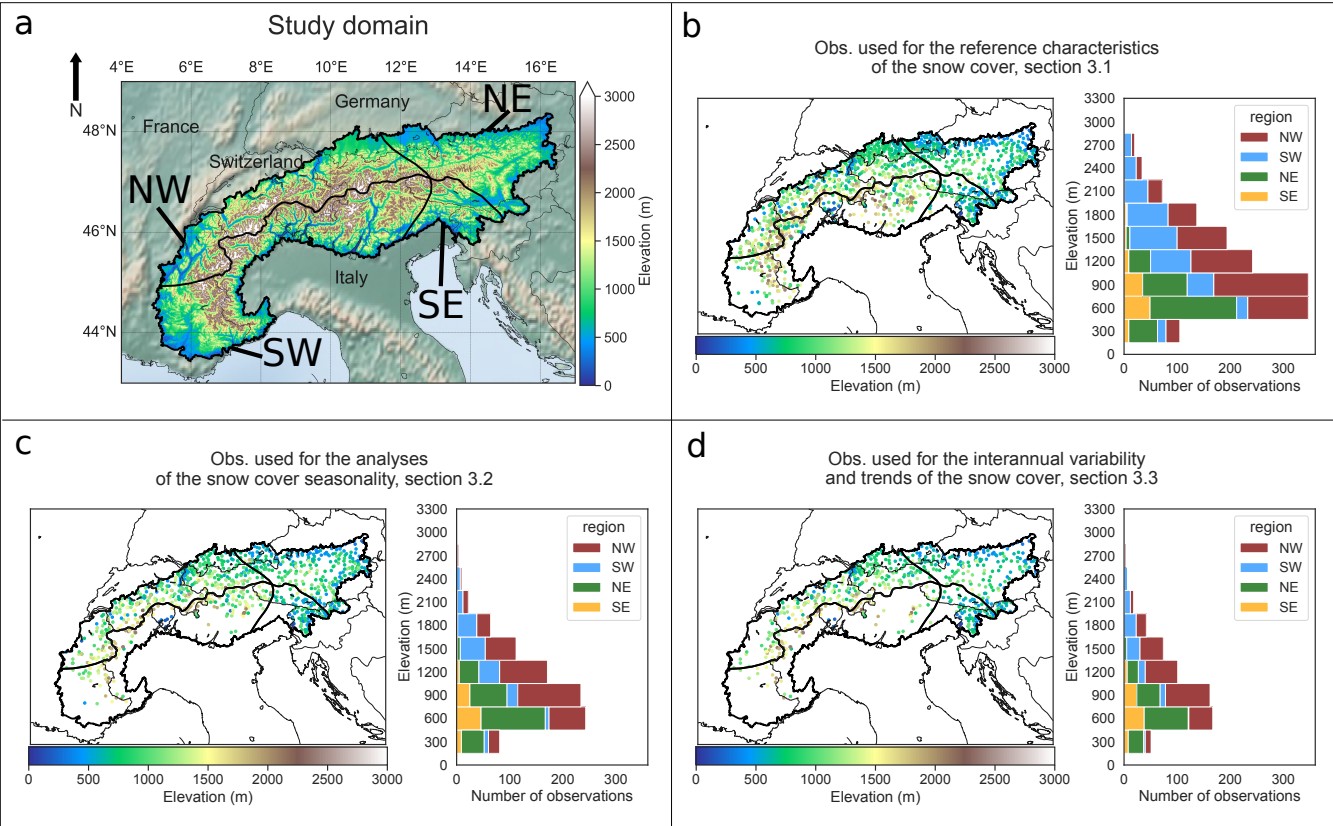

**Figure 1.** a. Study domain with the DEM (digital elevation model) at 1 km and contour of the Alpine Convention outline and the four subregions, b. Location of observations used in section 3.1 with their associated number per elevation bands of 300 m width for each regions, c. Same as b. for the observations used in section 3.2, d. Same as b. for the observations used in section 3.3.



## 2.2 Variables of interest and indicators

Based on the availability of the variables across the data sets used in this work (see below), we focus on snow depth to evalate their snow cover component.

In oder to compare the evaluated datasets against remote sensing data from MODIS/Terra (MOD10A1F) processed by the National Snow and Ice Data Center (NSIDC) (Hall and Riggs., 2020), three indicators are analysed. Consecutive snow cover duration (SCD) is defined as the largest consecutive number of days in a hydrological year with a snow depth value above a

given threshold value. The snow onset date (SOD) and the snow melt-out date (SMOD) characterize the beginning and end dates of the corresponding time period. In this study, the snow depth threshold is set at 1 cm, motivated by the minimization of error metrics, described in Section 2.4.3.

In the case of MODIS data, the normalized difference snow index (NDSI) value was converted to a series of binary snow cover maps (absence or presence of snow) using a threshold value NDSI > 0.2. This threshold corresponds to a snow cover

fraction of approximately 30% (Salomonson and Appel, 2004). These snow cover maps were used to compute SCD, SOD and SMOD.

Further to the state of the snow cover, we focus on near surface temperature (2 m temperature) and precipitation amounts, at the seasonal scale relevant to the winter snow cover (average and cumulated values from November to April, respectively).

## 2.3 Data

### 2.3.1 Reference datasets

**In-situ snow depth observation**

The reference snow depth dataset is an ensemble of daily in-situ observations spanning the 1971-2019 period (Matiu et al., 2021). In this study, we used three different subsets from the overall dataset, depending on the analysis, with their locations and elevation distribution displayed in Figure 1. Section 3.1 focuses on the reference characteristics of the snow cover. In order

to use the largest possible spatial coverage to compare mean values of snow depth over large regions, all the observations available for the 1985-2015 time period are included (see Figure 1b). Section 3.2 focuses on the snow cover seasonality, using the indicators SCD, SOD and SMOD computed using continuous daily values of snow depth over the winter period. Most of the missing values are concentrated in summer (for most of the observation stations, no snow is present during this period). We removed all stations with more than 20% of missing values over the 2000-2015 period (see Figure 1c). Section 3.3 focuses

on the interannual variability and trends. This requires the most homogeneous possible dataset, so that we remove all stations with more than 10% of missing values over the winter month (November to April) of the 1968-2018 period (see Figure 1d).

**MODIS remote sensing satellite observations**

In order to address the spatial variability and the snow cover seasonality, we used the MODIS/Terra daily normalized difference snow index (NSDI) field over the 2000-2015 period. These data from the MODIS/Terra sensor have been treated by the National





Snow and Ice Data Center (NSIDC) (Hall and Riggs., 2020) and it corresponds to a daily gap-filled product using an algorithm described in Hall et al. (2010). In this study, MODIS NDSI data are used at their native horizontal resolution ( 500 m), and regridded to match different dataset resolutions (from 2.5 km to 30 km) using a conservative method.

**Near-surface air temperature**

The reference air temperature at 2 meters dataset is the E-OBS v23.1 daily mean air temperature field (Cornes et al., 2018). E-
OBS is a gridded observational dataset available at 0.1° ( 12 km) horizontal resolution over the 1950-2020 period. It is obtained by interpolating station data gathered from National Meteorological Services (NMSs) by the ECA&D initiative (Klein Tank et al., 2002; Klok and Klein Tank, 2009). Uncertainties (due to climatological standard errors value and from the kriging procedure) are estimated using stochastic simulations to produce en ensemble of 100 realizations of each daily fields, then the spread is calculated using the 95th and 5th percentile. In section 3.3, focusing on interannual variability and trends, we used
the homogenized version v19.0HOM of E-OBS. It uses a restricted number of observations, quality checked and homogenized following a procedure described in (Squintu et al., 2019). E-OBS temperature values in high mountain areas present limitations. They are known to feature a high warm bias (that can reach up to 5°C against the Meteoswiss gridded dataset in Switzerland), resulting from the scarcity of observations used in this region at high elevation. Furthermore, the uncertainties values may be underestimated, particularly in areas with low observation density (Cornes et al., 2018).

**Precipitation**

The reference for precipitation is LAPrec (Frei and Isotta, 2019), a gridded dataset of monthly precipitation at 5 km horizontal resolution covering the European Alps and spanning the 1901-2019 period. It relies on a statistical approach (Reduced Space Optimal Interpolation - RSOI) combining information from a set of long-term observation stations used in HISTALP (Auer et al., 2007) and from the high resolution gridded datasets EURO4M-APGD (Isotta et al., 2015). It is specifically appropriate
for long-term studies that need a high temporal consistency while staying at a relatively large spatial scale, therefore matching the scope of this study. The user guide (Isotta et al., 2021) of this product provides an estimation of the interpolation error (in terms of mean absolute error - MAE and ME) at observation locations. The value of the MAE is 18 mm/months (all stations and months included), but is found to be larger in areas with a lower density of observations (i.e. generally at high elevation), and in summer, due to the larger proportion of convective precipitation events. Additionally, systematic error measurements
are not taken into account, such as the rain gauge undercatch due to wind-induced deflections of hydrometeors, known to be particularly strong at high elevations in winter (i.e. the underestimation can occasionally exceed 40%) (Isotta et al., 2021; Kochendorfer et al., 2018, 2021).

### 2.3.2 Evaluated dataset

Reanalyses and climate simulations are evaluated in this study against the references described above. We however emphasize
that these references are not immune of errors and serve as a common reference dataset for the purpose of this work. Figure





2 shows an overview of the evaluated datasets, their configurations, temporal availability, as well as their main components of interest for this study (i.e. land surface model and complexity of their snow scheme).





**Figure 2.** Schematic description of the evaluated datasets. Each dataset is represented in a colored rectangle with a width adjusted to fit their temporal coverage on the timeline. A one-way arrow indicates a driving element that is used inside an other model in stand-alone mode (e.g. global driving data for regional model, atmospherical field for offline LSM run). A two-ways arrow indicates a coupling between a driving element and an other model. Dashed squares gives specifications on their associated element.



### 2.3.3 Reanalyses

**ERA5**

ERA5 is the latest global reanalysis produced by the ECMWF using the Cy41r2 of the Integrated Forecasting System (IFS). This reanalysis provides hourly atmospherical and surface fields at a horizontal resolution of 31 km. Here we use the latest release, at the time of writing, of the reanalysis starting in 1959, while a previous version of the reanalysis was produced, starting in 1950, and is used for ERA5-Land and ERA5-Crocus, see below. A 4D-Var assimilation framework including variational bias correction is used for atmospherical fields, 2D optimal interpolation for 2 m temperature, relative humidity and snow (depth

and density), and 1D Optimal Interpolation for soil and snow temperature (Hersbach et al., 2020). The Land Surface Model (LSM) CHTESSEL integrates a single explicit layer snow model. It is an energy and mass balance model that represents an additional layer on top of the upper soil layer (Dutra et al., 2010), with its own energy budget, taking into heat exchanges with the underlying soil and above atmosphere. It has a comparable physics as the D95 snow cover model (Douville et al., 1995), but accounts for more processes : the representation of liquid water content (as a diagnostic) and the compaction and thermal

metamorphism in its snow density formulation (see Dutra et al. (2010) for more details). It is worth noting that some issues affect the ERA5 reanalysis snow depth data. Hersbach et al. (2020) note that above 1500 m in mountainous area, snow depth can be unrealistically large, due to the underestimation of melting within the snow scheme parameterization.

**ERA5-Crocus**

ERA5-Crocus corresponds to driving the LSM SURFEX (Masson et al., 2013) used in standalone mode along with the detailed

multi-layer snowpack model Crocus (Vionnet et al., 2012), using as input the meteorological fields from the ERA5 reanalysis at 31 km horizontal resolution covering the 1950-2020 period, over the Northern hemisphere. ERA5-Crocus has been used in several recent analyses of northern-hemisphere snow cover trend (Derksen and Mudryk, 2022).

**ERA5-Land**

ERA5-Land is a global reanalysis produced by the ECMWF for the land component from 1950 onwards at a horizontal

resolution of 9 km. It uses the ERA5 atmospherical fields downscaled at 9 km of resolution using a linear interpolation with an altitudinal correction for the air temperature, humidity and pressure. The altitudinal correction is achieved using a daily environmental lapse rate derived from ERA5 vertical profiles (Dutra et al., 2020). We note that Dutra et al. (2020) only shows nuanced benefits of this altitudinal correction on temperature over the western US, with even a deepening of a cold bias against station observations when ERA5 elevation is corrected towards higher elevation, the dominant situation over high mountain

range. These downscaled atmospherical fields are then used to force the LSM CHTESSEL (using a similar configuation as the one used in the ERA5 reanalysis), producing hourly surface fields.



**UERRA : MESCAN-SURFEX**

UERRA was a European project focused on the development of regional-scale atmospheric and land surface reanalyses. Multiple datasets were produced within the framework of the UERRA project. Here we used the MESCAN-SURFEX analysed fields. It is a regional reanalysis covering Europe and spanning the 1961-2019 period. It provides analysed near-surface atmospherical and surface fields each 6 hours at a horizontal resolution of 5.5 km, and forecast fields hourly. It uses ERA-40 (Uppala et al., 2005) prior to 1979 and ERA-interim (Dee et al., 2011) thereafter as lateral boundary conditions (global forcing) to run the HARMONIE (HIRLAM ALADIN Regional/Mesoscale Operational NWP In Europe) NWP system at 11 km of resolution (Ridal et al., 2016). An analysis is done every 6 hours using a 3D-Var assimilation for the upper atmosphere and CANARI (Taillefer, 2002) for the surface. The analysed atmospherical fields at 11km horizontal resolution are downscaled to 5.5km and passed to the MESCAN system (Bazile et al., 2017; Soci et al., 2016), producing an analysis of the air temperature at 2 m, the humidity at 2 m as well as the precipitation. These surface fields along with radiation and wind fields from the atmospherical analysis are used to drive the SURFEX LSM in standalone mode to produce surface fields at 5.5km grid spacing. For this study, precipitation and air temperature correspond to the MESCAN analysed fields, and snow depth values are produced by the LSM SURFEX used in standalone mode. In this configuration SURFEX uses the intermediate complexity 12 layers snow scheme ISBA-ES (Explicit Snow) (Boone and Etchevers, 2001; Decharme et al., 2016).

**MTMSI**

The Mountain Tourism Meteorological and Snow Indicators (MTMSI) correspond to a a series of indicators generated at the pan-European scale based on a selection of grid points from the UERRA-MESCAN-SURFEX meteorological reanalysis over a specific geometry (elevation bands every 100 m elevation within each mountainous NUTS3 area) used as inputs of the detailed snowpack model Crocus, from 1961 to 2015. Temperature and precipitation fields are directly derived from the MESCAN analysis, while snow depth is produced by the snowpack model Crocus. See Morin et al. (2021) for further details about this dataset.

**CERRA-Land**

CERRA-Land is the latest generation regional reanalysis covering Europe from 1984 onwards, produced as part of the Copernicus Climate Change Service (Schimanke et al., 2022). It provides near surface atmospherical and surface analysed fields each 3 hours at an horizontal resolution of 5.5 km. Its setup is almost similar to UERRA-MESCAN-SURFEX. The main differences are the use of ERA5 as lateral boundary conditions, the fact that the atmospherical model (HARMONIE) runs natively at 5.5 km of horizontal resolution and that an analysis takes place every 3 hours.



### 2.3.4 Climate simulations

**CNRM-ALADIN**

The CNRM-ALADINv6 regional climate model (Nabat et al., 2020) uses a 12.5 km horizontal grid spacing over a large pan-European domain, 91 vertical levels and a 450 s internal time step. It is hydrostatic, which involves the parametrisation of deep convection, using the PCMT (Prognostic Condensates Microphysics and Transport) scheme (Piriou and Guérémy, 2016). The coupling with the LSM SURFEX includes the snow cover model ISBA-ES, using a 12-layers snowpack discretisation scheme. Here, we used an evaluation run spanning the 1979-2018 period, using ERA-interim as its lateral boundaries.

**CNRM-AROME**

This study relies on simulations carried out with CNRM-AROME (cycle 41t1) at 2.5 km horizontal grid spacing (Caillaud et al., 2021; Monteiro et al., 2022). This version of the model was the one used operationally for NWP at Météo-France from 2015 to 2018 (Termonia et al., 2018). CNRM-AROME includes a coupling with the LSM SURFEX, using the single layer D95 snow scheme (Douville et al., 1995). Here, we used an evaluation run spanning the 1982-2018 period, that used CNRM-ALADIN evaluation run as lateral boundary conditions (Monteiro et al., 2022) .

### 2.4 Evaluation methods

#### 2.4.1 Regional averaged analyses

In order to provide a common basis for the evaluation of these diverse datasets, we aggregate the temperature, precipitation and snow cover data for relatively large areas (full European Alps domain, or subregions) and for given elevation bands.

#### 2.4.2 Elevation band based analyses

Figure 3a displays the relative frequency of the number of grid points for a digital elevation model (DEM) at 100 m horizontal resolution by elevation band and region, and Figure 3b, the difference in terms of the number of grid points for each dataset topography compared to the reference. It highlights that the elevation distribution over the subregions presents strong discrepancies for the different datasets investigated. Moreover, mountainous regions are known to feature a large altitudinal gradient concerning the variables of interest for this study, at least for the snow cover state and near surface air temperature. Comparing multiple datasets at different resolutions without taking into account this unequal repartition of grid points per elevation band would inevitably induce strong systematic biases in the analysis results. Here, the analyses are carried out using averages over 300 m-witdh elevation bands, meaning that for a given elevation band of elevation $z$, all stations/grid points with an elevation ranging between $z\pm150$ m elevation group are combined (usually through averaging).This choice results from a trade-off between the heterogeneity within an elevation band for a given region, and the inclusion of a maximum of grid points or observations within.




For most of the analyses, we chose to focus on three elevation bands, acting as a representation of three distinct environment
: 600 m±150 m for the valleys and low elevation hills, 1500 m±150 m for the intermediate elevation near the snow line and
2400 m±150 m for the high mountain conditions.

Last section includes figures with average over multiple elevation bands, where we exclude data below 450 m since snow has
a very limited role at these elevations and above 2550 m as our used datasets are not designed to represent very high elevations
conditions. Overall, it leads to the exclusion of less than 12% of the total surface based on the DEM at 100 m resolution (see
Figure 3a).

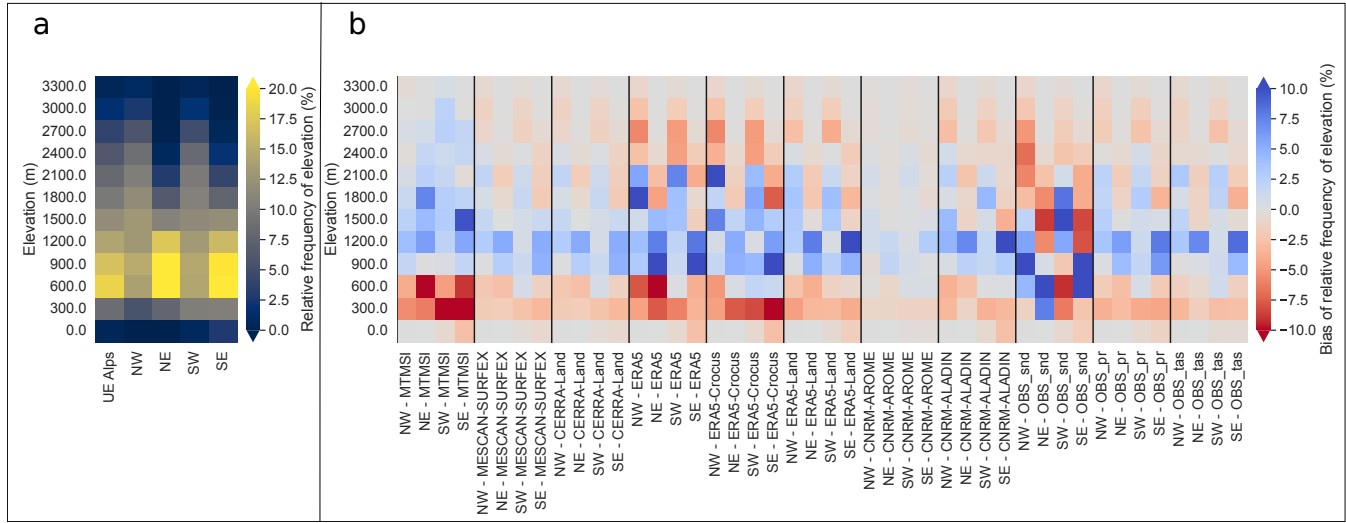

**Figure 3.** a. Relative frequency of the number of grid points for 300 m width elevation bands for each regions using a digital elevation model
at 100 m horizontal resolution. b. Differences in the frequency of the number of stations or grid points for 300 m width elevation bands for
each regions and datasets.

### 2.4.3 Determination of the snow depth threshold

The computation of the snow cover duration (SCD), snow onset date (SOD) and snow melt-out date (SMOD) requires as a
first step to convert any snow data (e.g. snow depth, snow water equivalent, NDSI ... ) into a binary data, informing about
the presence or the absence of snow. As it does not exist a consensus for a snow depth threshold to determine the absence
or presence of snow at a given location, we choose the threshold that maximizes the agreement of the snow cover detection
between MODIS (with a NDSI threshold set at 0.2 to correspond to approximately 30% of the pixel snow-covered (Salomonson
and Appel, 2004)) and in-situ station observations (with a varying threshold of snow depth), as well as minimizing error metrics
on the used indicators (SCD, SOD and SMOD). The agreement metrics used are skill scores based on the confusion matrices
calculated using daily values of presence or absence of snow, considering in situ observations as the truth : the true positive
rate (i.e. number of points flagged as positive in MODIS pixel and corresponding station - TPR), the true negative rate (i.e.
number of points flagged as negative in MODIS pixel and corresponding station - TNR), the false positive rate (i.e. number





of points flagged as negative in MODIS pixel but positive in the corresponding station - FPR), and the false negative rate (i.e. number of points flagged as positive in MODIS pixel but negative in the corresponding station - FNR). Two additional scores incorporating both information on agreement and disagreement are calculated : the accuracy (i.e. the proportion of the total
number of predictions that were correct) and the kappa coefficient (Cohen, 1960).

Then, differences for the different thresholds between MODIS and stations observations for our three indicators (SCD, SOD, MOD) are quantified using mean absolute error (MAE) and mean error (ME) values.

The stations used in section 3.2, described in Figure 1c have been chosen, because this subsets covers all the elevations of interest for this study with a satisfying spatial coverage. In order to avoid altitudinal biases, stations that presents elevation
differences greater than 100 m with their corresponding MODIS pixel have been removed, as well as stations below 450 m and above 2550 m (247 out of 941 stations have been removed). In total 694 stations over the period from April 1st, 2000 to December 31st, 2015 have been used, providing 3'886'400 daily observations for the calculation of skill scores, and 14 seasons (2000-2001 to 2014-2015) for the SCD, SOD and SMOD MAE and biases.

Figure 4a shows that the threshold that maximize the kappa and the accuracy between MODIS and observation stations is
1 cm for the detection of snow covered pixels. This threshold is lower than the 10 cm to 15 cm optimum found by Gascoin et al. (2015) for the Pyrenees. Possible factors that could explain the differences are a larger number of observations used in our study, covering a larger area with a larger number of low elevation sites, and a wider range of land-cover. Giving a clear explanation of the differences would require further sensitivity studies that are beyond the scope of this study.

Figure 4b show errors metrics for the different threshold, with shaded areas representing the standard deviations between the
elevation classes (from 600 m±150 m to 2400 m±150 m each 300 m). SOD appears to be weakly sensitive to this threshold, meaning that the beginning of the continuously snow-covered season generally starts with high values of snow depth. SMOD and SCD are more sensitive, and error metrics grown continuously with the increase of the threshold. The standard deviation is rather low for the 1 cm threshold (i.e. less than 5 days for mean error and mean absolute error), meaning that the detection performs rather similarly at all elevations. This short sensitivity study leads us to the choice of 1 cm for setting the snow depth
threshold used to determine the absence or presence of snow with an uncertainty that can be large for a given station (i.e. on average 10 to 20 days), but well centered as mean error values are close to 0. This threshold was applied to all the datasets compared with MODIS in section 3.2, regardless of their horizontal resolutions, which can be considered as a limitation of the method.



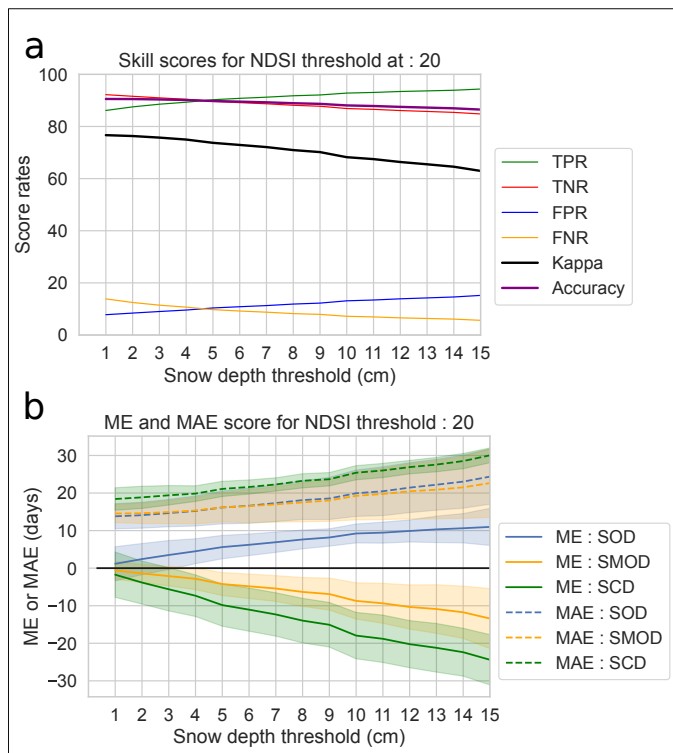

**Figure 4.** a. Skill scores (true positive rate or recall - TPR, true negative rate - TNR, false positive rate - FPR, false negative rate - FNR, accuracy and kappa), calculated using the daily values of absence or presence of snow on the ground between in-situ stations and MODIS pixels at station locations, for various values of the threshold used to assess whether snow is present or not. See text for more details. b. Corresponding mean errors and mean absolute errors for the SOD, SMOD and SCD indicators. Shaded areas represent the standard deviations between the elevation classes (from $600\,\mathrm{m}\pm150\,\mathrm{m}$ to $2400\,\mathrm{m}\pm150\,\mathrm{m}$ each $300\,\mathrm{m}$)

#### 2.4.4 Time periods, statistics and trends analyses

The reference period for the analyses in section 3.1 is the longest common period available for all datasets : 1985-2015 (see Figure 2). Section 3.2 that focuses on snow cover seasonality mainly against MODIS data is performed over the 2000-2015 period. In these two sections, most values of the different variables are presented on average over the time period, spatially averaged for a given elevation band, over a subregion or the whole Alps.

   The error metrics used are defined as follows :

– Mean error (ME) : $ME = \frac{\sum_{i=1}^{N}(x_i - y_i)}{N}$

   – Mean absolute error (MAE) : $MAE = \frac{\sum_{i=1}^{N}|x_i - y_i|}{N}$

   – Root mean square error (RMSE) : $RMSE = \sqrt{\frac{\sum_{i=1}^{N}(x_i - y_i)^2}{N}} = \sqrt{1 - r^2}\sigma_y$

   – Correlation (Pearson linear correlation) : $r_{xy} = \frac{\sum x_i y_i - n\bar{x}\bar{y}}{\sqrt{(\sum x_i^2 - n\bar{x}^2)}\sqrt{(\sum y_i^2 - n\bar{y}^2)}}$



with $x_i$ and $y_i$, data $x$ and $y$ at time step $i$, $\sigma_x$ et $\sigma_y$ respectively the standard deviation of $x$ and $y$ and $N$ the sample size.

For convenience, ME, MAE and RMSE are normalized by the mean values of the reference dataset for snow depth and precipitation.

Section 3.3 compares trends for the different variables of interest for the reference and evaluated datasets calculated using monthly mean winter values (November to April). As monthly snow depth time series are heteroscedastic (non uniform variance), trends are calculated using the robust nonparametric Theil-Sen estimator, insensitive to the changing variance of

the residuals (Sen, 1968). The same analyses have been performed using standard ordinary least squares regressions (OLS), and generalized least squares (GLS) with an autoregressive component (AR(1)) to account for the effect of the interannual variability on the calculated trends, known to lead to an increase of the confidence interval amplitudes (Ribes et al., 2016). The resulting trends are of comparable amplitudes for the three methods, but the OLS method leads to the detection of more significant trends, compared to GLS with AR(1) and Theil-Sen methods, which generally share similar thresholds significance

levels for our analysed time series.

## 3    Results

### 3.1    Reference characteristics of the snow cover in the European Alps

This section presents a general evaluation of snow depth characteristics, air temperature at 2 meters and precipitation for each of the datasets by comparing them against the reference datasets presented in section 3.1 for the 1985-2015 period. Comparisons

are described at the scale of the whole Alps, there are not much differences in the results at the sub-regional scale. Figures for the four subregions are however provided in the supplementary materials.

#### 3.1.1    Snow depth

Figure 5 shows an overview of the snow depth monthly time series for the different datasets over the whole Alpine ridge. The corresponding values for the correlation, mean errors (ME) and mean absolute errors (MAE) compared to the reference

calculated using the monthly values for the snow season (November to April) are shown on Figure 6. On these two figures, three elevation bands are presented : 600 m (from 450 m to 750 m ; low elevation), 1500 m (from 1350 m to 1650 m ; intermediate elevation) and 2400 m (from 2250 m to 2550 m ; high elevation).

At all elevations on Figure 5, all datasets present similar interannual variability, reflecting a satisfactory agreement concerning the chronology of events (see section 3.3 for a more in-depth comparison of the interannual variability). This leads to high

correlation scores on Figure 6, ranging from 0.66 to 0.96, most of which above 0.85. At low elevation on Figure 6a, CNRM-ALADIN and ERA5 presents the lower correlation score of 0.66 and 0.70 respectively, mostly explains by a shifted timing of the snow accumulation and melting phase than the reference. Their relative mean error are negative (-75% for CNRM-ALADIN and -31% for ERA5) due to a too thin snow cover (see Figure 5a), a behaviour that is also found for the other datasets with the exception of ERA5-Land that overestimates the amount of snow, with a normalized mean error of +35%. At intermediate and



high elevations, climate simulations (CNRM-ALADIN and CNRM-AROME) and ERA5-Land systematically overestimate peak winter monthly snow depth values (see Figure 5bc and Figure 6.a ), leading to a relative mean error between 49% and 85% at intermediate elevation (81% to 191% at high elevation). Figure 5c shows that at high elevation, snow depth values never drop below a given value for some products (0.2/0.3 m for CNRM-ALADIN and CNRM-AROME, 1.8 m for ERA5-Land). MESCAN-SURFEX, MTMSI, CERRA, ERA5 and ERA5-Crocus display the lowest mean errors, with relative MAE
values rarely exceeding 30% as well as correlation values always close to 0.9 for every elevation bands.







**Figure 5.** Time series of monthly snow depth values for each dataset for three elevation bands for the available common period, 1985-2015 over the entire European Alps domain. The reference OBS (in-situ observations of snow depth) is in black line with circle markers. a - Time series for the elevation band 600 m ±150 m, b - Time series for the elevation band 1500 m ±150 m, c - Time series for the elevation band 2400 m ±150 m.





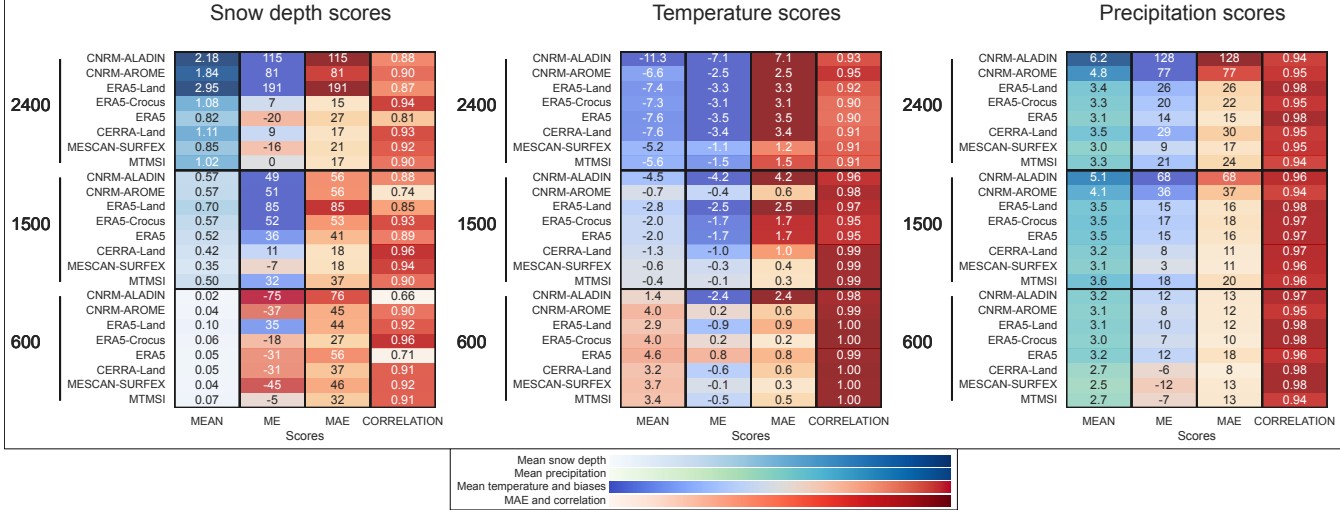

**Figure 6.** Mean values and scores of mean errors (ME), mean absolute errors (MAE) and correlations calculated using mean monthly values from November to April over the 1985-2015 period for the whole Alps and compared to the reference (in-situ observations dataset for the snow depth, E-OBS for the temperature and LAPrec for the precipitation) for each dataset for three elevation bands (600 m ±150 m, 1500 m ±150 m, 2400 m ±150 m). Scores are presented for snow depth, temperature and precipitation, and scores of MAE and biases are normalized by the mean values of the reference (given in %) for snow depth and precipitation.

Figure 7abc shows, in the form of boxplots for three elevation bands, the distribution of the mean snow depth values from November to April for each dataset. Figure 7def shows the corresponding annual cycle. It confirms the previous analysis, and allows a more in-depth analysis of the behaviour of some datasets.

At all elevations, ERA5-Land overestimates the amount of snow. In winter, snow depth values are about twice larger than the reference values at low and high elevations, and around 50% higher at intermediate elevation. It also leads to a snowpack lasting too long until melt, particularly at intermediate and high elevations, reaching its peak winter values and with a beginning of the melting period one month later than the reference.

CNRM-AROME also overestimates the snow depth and the duration of the snow cover at intermediate and high elevations. In this case, it is combined with a delayed accumulation phase depicted by an underestimation of the snow depth until it reaches its peak winter value. CNRM-ALADIN also exhibits an overestimation of the snow depth at intermediate and high elevations, but with a reversed behaviour concerning the accumulation timing. It overestimates snow accumulation at the beginning of the season with an earlier snow onset date, but starts melting the snowpack one month earlier than the reference, similar to what was found by Monteiro et al. (2022) in the French Alps.

The amplitude and timing of the beginning, peak and end of the season are close to the reference for MESCAN-SURFEX, CERRA and MTMSI, with discrepancies concerning the snow depth, within 20% during the season. It is not surprising that only subtle differences can be seen between these three datasets, as MTMSI is a selection of MESCAN-SURFEX grid points



(although with a different snow cover model, and MESCAN-SURFEX and CERRA reanalyses roughly share the same mod-elling systems.

The same conclusions than for MESCAN-SURFEX, CERRA and MTMSI can generally be reached for ERA5 and ERA5-
Crocus. They both present a satisfying average monthly evolution of the snowpack over the Alps, with differences against the reference that do not exceed 25%, and no strong deviations concerning the timing of the accumulation, peak and melting phase of the snow season. Neither of the two outperforms the other in terms of mean values, ERA5 showing alternatively less and more differences than the reference compared to ERA5-Crocus. Nonetheless, Figure 7bcd show that the use of Crocus offline drives by the atmospherical analysis of ERA5 improves slightly the monthly evolution of snow depth, leading to values closer
to the reference during the accumulation and melting time periods (this is also seen for individual subregions, see appendix A Figure A1).

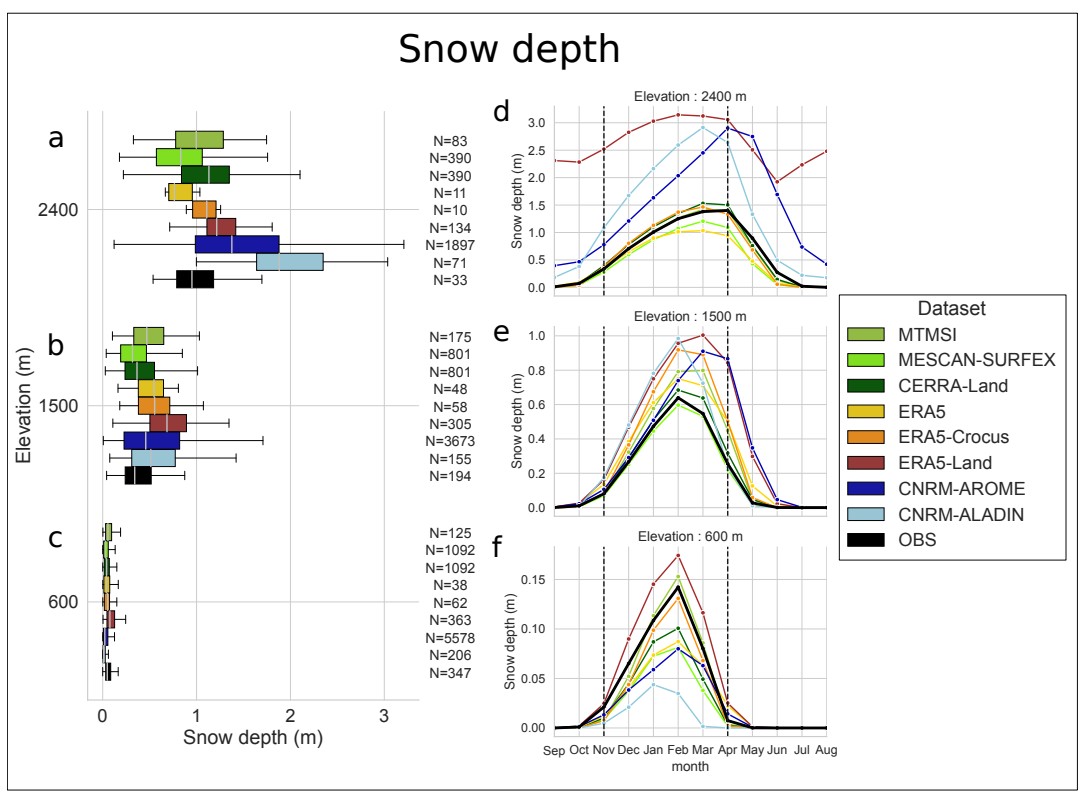

**Figure 7.** a, b, c - Boxplot representing the spatial distribution of mean winter season (November to April) snow depth values over the 1985-2015 period for each dataset for three elevation bands (600 m ±150 m, 1500 m ±150 m, 2400 m ±150 m) over the entire European Alps domain. d, e, f - Annual cycle of the mean monthly snow depth values for the 1985-2015 period for each dataset, for three elevation bands (600 m ±150 m, 1500 m ±150 m, 2400 m ±150 m).



### 3.1.2 Snow driving variables : temperature and precipitation

The following subsections are dedicated to the evaluation of temperature and precipitation, following similar analyses as for snow depth. Because temperature and precipitation can be considered as the first order driving variable influencing the state
of the snow cover, we also aim at identifying features that could be related to snow depth specific behaviour described before. Figure 8 shows similar results as Figure 7, but for temperature and precipitation.

**Air temperature at 2 meters**

At low elevations (Figure 8cf), most of the datasets do not exhibit significant differences with respect to the reference (i.e. outside the uncertainties range of E-OBS, $\pm$ 1°C), except climate simulations (CNRM-AROME and CNRM-ALADIN), presenting
similar patterns of deviations against the reference. CNRM-AROME temperature values are 1°C higher than the reference during the winter, and CNRM-ALADIN temperature values are 3°C lower at most during spring season. These features, displayed here at the scale of the European Alps, have already been identified in the French Alps (Arnould et al., 2021; Monteiro et al., 2022).

Intermediate and high elevation bands on Figure 8abde show more contrasting results. At intermediate elevation CNRM-
ALADIN and CNRM-AROME show larger discrepancies compared to E-OBS, with similar patterns overall. CNRM-AROME differences reach -1.8°C during spring, -5°C for CNRM-ALADIN in February. For ERA5-based products (ERA5, ERA5-Crocus and ERA5-Land), negative temperature differences are found and range from -2°C to -3.5°C from November to February, peaking in December and January. The strongest discrepancies are found between ERA5-Land and the reference dataset. At this elevation, MESCAN-SURFEX, MTMSI and CERRA, negative temperature differences of 1°C at most are also found
for the winter months but remain within the E-OBS uncertainty range. At high elevations, there are larger differences for all datasets. Similar to low and intermediate elevations, climate simulations (CNRM-ALADIN and CNRM-AROME) exhibit negative deviations to E-OBS. The strongest differences are in winter, peaking from December through February with CNRM-ALADIN temperature values being 7°C lower than E-OBS, CNRM-AROME 2.5°C lower. ERA5-based products show larger discrepancies compared to the intermediate elevation band, showing negative temperature differences around -5°C with respect
to E-OBS. At this elevation, MESCAN-SURFEX, MTMSI and CERRA also display negative temperature differences for the winter months with MESCAN-SURFEX differences with E-OBS around -2°C and CERRA differences reaching -4°C.

For the intermediate and high elevation comparisons, we need to remain cognizant of the the questionable reliability of E-OBS at these elevations (see Section 2.3.1). The negative temperature difference bwteen all datasets and E-OBS could indeed be due to a bias of E-OBS itself. Indeed Cornes et al. (2018) show that E-OBS can exhibit higher temperature values, up
to 5°C at high elevation locations, against the Meteoswiss reference temperature dataset (Frei, 2014). Nonetheless, part of these differences could also reflect a genuine negative temperature bias in the evaluated dataset as it as already been shown in previous studies. Dutra et al. (2020) showed that ERA5 and ERA5-Land present a median negative temperature bias of -1°C in winter in the Western US against a large number of in-situ observations, including the Rockies mountains. Monteiro et al.




(2022) showed that over the French Alps, CNRM-AROME and CNRM-ALADIN also exhibit a negative temperature bias in
winter at high elevation against the S2M reanalysis.

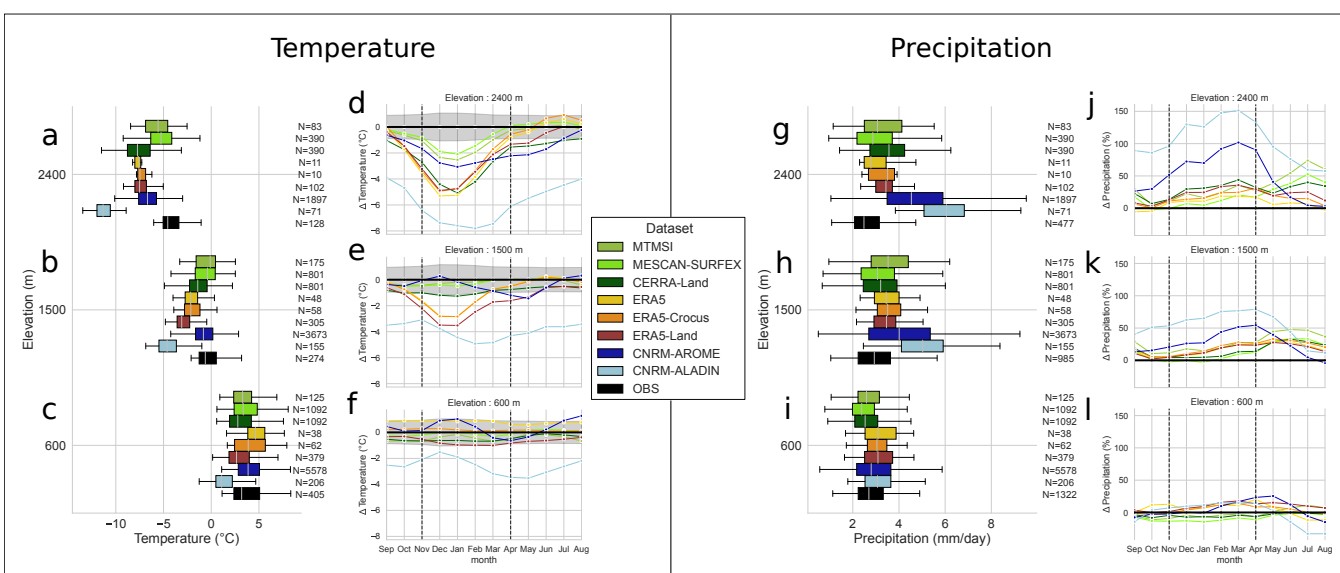

**Figure 8.** a, b, c and g, h, i - Boxplots representing the spatial distribution of mean winter (November to April) values of temperature and
precipitation over the 1985-2015 period for each dataset for three elevation bands (600 m ±150 m, 1500 m ±150 m, 2400 m ±150 m) for the
entire European Alps domain. d, e, f and j, k, l - Annual cycle of the mean monthly differences of temperature and precipitation compared to
the corresponding reference (E-OBS for temperature, LAPrec for precipitation) for the 1985-2015 period for each dataset, for three elevation
bands (600 m ±150 m, 1500 m ±150 m, 2400 m ±150 m). For the temperature, the shaded area represent the uncertainties associated with
E-OBS dataset.

**Precipitation**

Figure 8l shows that at low elevation, rather small differences can be seen between the datasets and the reference, ranging
between ±25% of the reference precipitation amount over the winter period (Novembre to April). The boxplot on Figure 8i
confirms that the winter distribution of the mean values at low elevation for the different dataset stay close to the reference,
with slightly higher precipitation values for ERA5-derived datasets and climate simulations (CNRM-AROME and CNRM-
ALADIN). MESCAN-SURFEX, CERRA and MTMSI values are close to LAPrec distribution and mean values (Figures 8il).
      At intermediate and high elevations, Figures 8ghjk show that all datasets indicate a higher amount of precipitation than
LAPrec, particularly during the winter season. Apart from climate simulations CNRM-AROME and CNRM-ALADIN, differ-
ences in monthly values range between 10% and 30% at the beginning of winter (November) and slightly increase along the
winter to peak in March, ranging from 25% to 50% at most, with strongest discrepancies always found for ERA5-Land and
CERRA. Climate simulations strongly exceed LAPrec values by 50% and 100% for CNRM-AROME to 80% and 150% for
CNRM-ALADIN at intermediate and high elevation, respectively, with largest differences at the end of winter (February to





April). Figure 8ghi confirms that the spread in winter precipitation values, and difference to the reference dataset, increase with elevation.

Although winter precipitation differences are consistent among regions (not shown here, see appendix A Figure A3) and datasets, summer precipitation values display more contrasted results with no clear pattern of over- or underestimation, neither among regions nor among elevation bands.

Overall, excepted the climate simulations CNRM-AROME and CNRM-ALADIN, the monthly values of precipitation for the different datasets remains rather close to the LAPrec dataset. Our results are in line with Bandhauer et al. (2022), who
compared ERA5, E-OBS and APGD over the Alpine region. They found an overestimation of 15-20% of ERA5 winter precipitation over LAPrec dataset. However, this overestimation, which we also identify for all the other datasets during the winter period could be nuanced. Indeed, Isotta et al. (2021) indicate that the LAPrec framework does not account for snowfall gauge undercatch. Bandhauer et al. (2022) estimated the underestimation to be close to 30% above 1500 m in winter, and Vionnet et al. (2019) showed that a regional reanalyses assimilating unshielded rain gauges can underestimates up to 50% of snow
mass accumulation over glaciers in the French Alps. We therefore expect that, except climate simulations CNRM-AROME and CNRM-ALADIN that lead to overestimated precipitation amounts above 1500 m, the overestimation found for the other datasets could rather be due to a more realistic estimate of the winter precipitation than LAPrec. Such results are also suggested from studies in North America (Wrzesien et al., 2019). We would however mention that if our results gives confidence to monthly and seasonal mean winter values of precipitation for most of the dataset, they do not provides information on the
temporal distribution of precipitation. For that point, it has for example been documented in Bandhauer et al. (2022) that ERA5 strongly overestimates the wet day frequency (up to a factor 2) over the Alps.

### 3.2 Timing of the snow cover duration against MODIS observations

In this section, we investigate further the timing of the snow season by comparing the evaluated datasets against remote sensing data from Terra/MODIS MOD10A1F (MODIS in the following) over the 2000-2015 period. The comparison is carried out
using indicators that can be either defined using the snow depth or the normalized difference snow index (NDSI) : the snow cover duration (SCD), the snow onset date (SOD) and the snow melt-out date (SMOD) (see sections 2.2 and 2.4.3 for further details). Figure 9ab shows maps and boxplots of the mean error between the evaluated datasets and MODIS. To this end, MODIS-based indicators were interpolated using conservative regridding over each dataset grid. This section does not include the MTMSI dataset because of its peculiar geometry (elevation bands over NUTS-3 areas). We also exclude CNRM-ALADIN
simulations because daily snow depth values are not available.

Figures 9a and b reveal positive differences across elevation bands, regions and datasets against MODIS, indicating a general overestimation of the snow season duration (SCD). Looking at the boxplot differences for the whole Alps (Figure 9b) brings more details about the elevational distribution of differences as well as their magnitude. All datasets display differences to MODIS values, which differ with elevation, peaking at intermediate elevation with median mean error (ME) ranging from 20
to 70 days.



The amplitude of the overestimation compared to MODIS is the strongest and the most spatially extended for ERA5-Land, exceeding 100 days in some locations (Figure 9a) with median mean error ranging from 40 to 75 days (Figure 9b). This confirms the issues with ERA5-Land snow cover data, consistent with the general overestimation of snow depth values and to the spurious snow accumulation occuring above 2000 m, shown in section 3.1, Figures 5, 6 and 7.

ERA5 and ERA5-Crocus data show lower mean error values than ERA5-Land, with largest differences in the western part of the Alps for ERA5. The median of the mean error values ranges from 20 to 50 days for ERA5, 10 to 40 days for ERA5-Crocus. While ERA5-Crocus does not show substantial improvements over ERA5 in terms of snow depth (see section 3.1.1), it provides results closer to the reference in terms of snow cover duration. MESCAN-SURFEX and CERRA snow cover duration data show similar patterns with either over and underestimation compared to MODIS. Indeed, Figure 9a shows an underestimation of the

SCD of 10 to 30 days (beyond 50 days at specific locations) over the inner alpine ridge, near the northwestern and southwestern boundary, and an overestimation elsewhere, ranging from 10 to 50 days. Their distribution of mean error values are overall more centered around 0 than the other datasets. MESCAN-SURFEX shows the lowest differences overall, and we note that CERRA particularly overestimates the SCD in the western Italian Alps.

For CNRM-AROME, the map on Figure 9a reveals an elevational pattern with an underestimation of the snow cover duration

at low elevation and an overestimation at higher elevation, consistent with the boxplot Figure 9b. This is in line with its snow depth underestimation at low elevation, overestimation at intermediate and high elevations described in section 3.1.



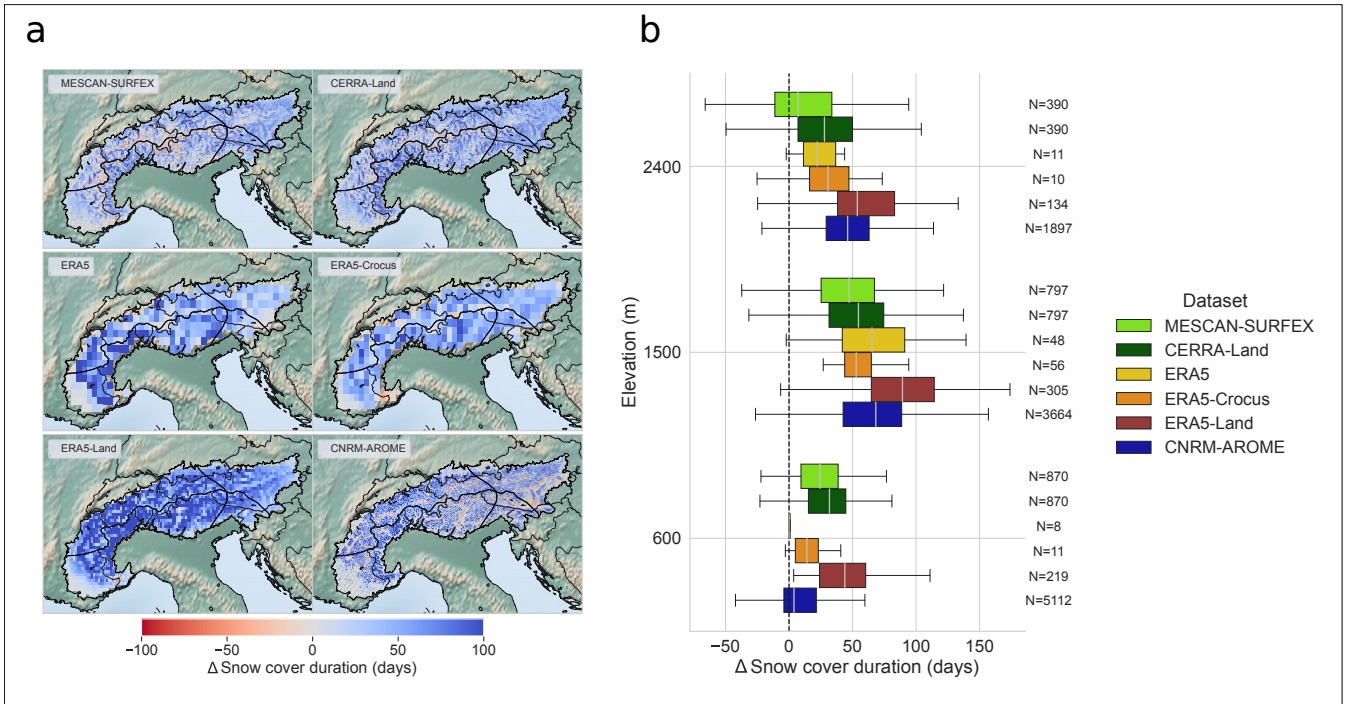

**Figure 9.** Snow cover duration differences between our evaluated datasets and MODIS observations in the European Alps. In order to compared MODIS to each datasets, MODIS at 500 m horizontal resolution have been reggrided using a conservative method over each dataset grid. a - Map of the average differences (mean error) of the snow cover duration (SCD) over 15 seasons (2000-2001 to 2014-2015) for each datasets compared to MODIS SCD. b - Boxplot representing the spatial distribution of the average differences (mean error) of the SCD over 8 seasons (2000-2001 to 2014-2015) compared to MODIS SCD for each datasets for multiple elevation bands.

Figure 10 provides information about the timing of the snow season by displaying the mean values of the snow onset date (SOD) and snow melt-out date (SMOD) as the edges of the barplot, while the barplot length corresponds to the snow cover duration (SCD). The error bar represent the spatial variability (using the standard deviation) of the SOD and SMOD for each dataset and MODIS at its native resolution.

Combining the information of Figures 9ab and 10abc shows that the snow cover duration is overestimated by all datasets, compared to MODIS, for all elevation bands. This also shows that the spatial variability of the SOD and the SMOD, represented by the size of the errorbars, are strongly underestimated in all datasets, excepted CNRM-AROME.

At low elevation (Figure 10a), MESCAN-SURFEX, CERRA and ERA5-Land provides SOD values close to MODIS, but SMOD values are too late up to 15 days, i.e. 30% longer SCD altogether. At this elevation ERA5-Crocus SOD and SMOD are rather close to the MODIS values, while ERA5 SOD occurs 15 days later, and CNRM-AROME SOD 7 days earlier, despite a close overall SCD. We note that in-situ observations also provide values towards a later beginning and longer lasting snow season, compared with MODIS estimates.





At intermediate elevation (Figure 10b), all datasets overestimate the duration of the snow season from 30 to 100 days at most, with an earlier SOD and a later SMOD compared to MODIS dataset. ERA5-Land and ERA5 results provide the longest SCD (around 170-200 days), while MESCAN-SURFEX, CERRA, ERA5-Crocus and CNRM-AROME provide SCD values, ranging from 135 to 150 days closer to MODIS values (about 100 days). At this elevation, the range of values of in-situ observations of SOD and SMOD are close to the range of values of MODIS, with an ealier mean of SOD and later mean of SMOD for the in situ observations that may be due to the oversampling of sites with a longer snow cover duration.

At high elevation (Figure 10c), whereas all datasets indicate a SOD value earlier than MODIS (around 15 days earlier), MESCAN-SURFEX, CERRA, ERA and ERA5-Crocus SMOD values are in agreement with MODIS values. ERA5-Land and CNRM-AROME SMOD values are 15 to 30 days later than MODIS-based estimates.

This comparison with MODIS indicators adds another perspective to Section 3.1: despite different behaviours concerning snow depth values at the monthly or seasonal scale, all datasets overestimate the duration of the snow season at all elevations compared to MODIS-based information. This is rather due to a late snow-melt out date, rather than an earlier snow onset date, with the largest discrepancies occuring at intermediate elevations. This relates to the fact that the snow melt-out date results from cumulated processes throughout the winter season, making it more difficult to estimate than other snow cover indicators.



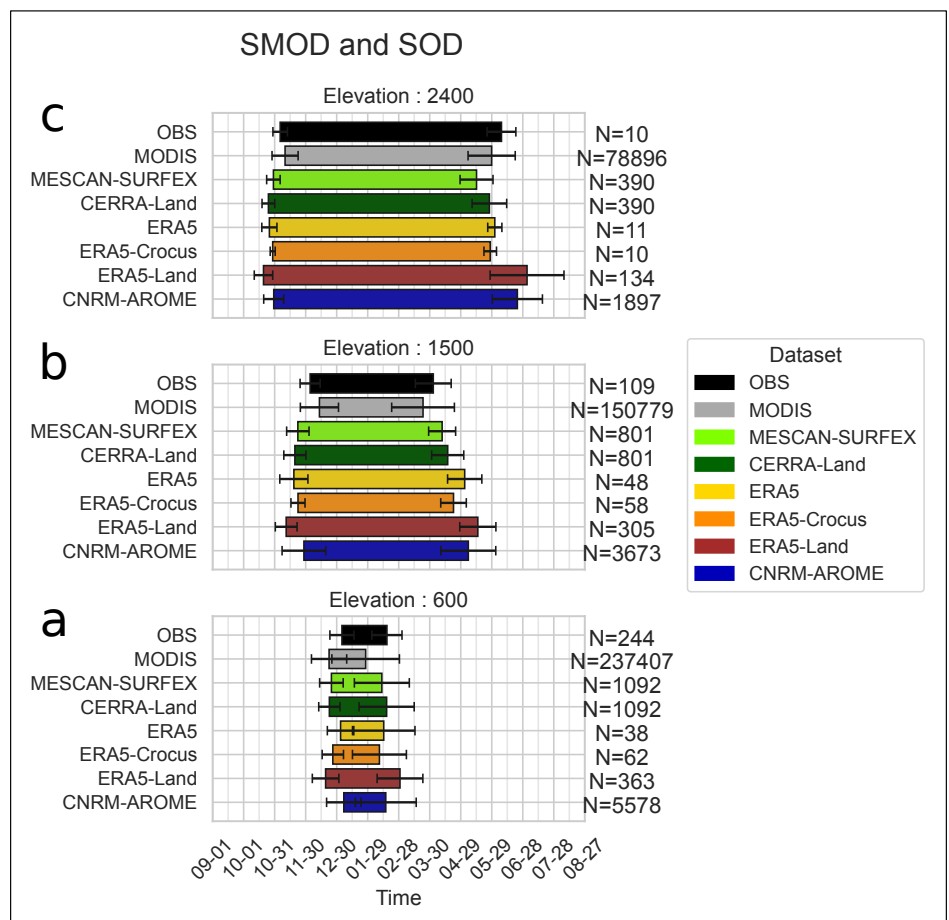

**Figure 10.** Barplot whose edges represent the spatial mean over the Alps of the snow onset date (SOD) and the snow melt-out date (SMOD) for 15 seasons (2000-2001 to 2014-2015) for three elevation bands (a - 600 m ±150 m, b - 1500 m ±150 m, c - 2400 m ±150 m) for each dataset at their native resolution. The errorbars surrounding the edges of each bar represent the standard deviation of the spatial distribution of mean values of the SOD and the SMOD.

### 3.3 Interannual variability and trends

In this section we analyze and compare the time variability and past trends for the winter season (November to April) obtained
using our reference and evaluated datasets over multiple time periods for most of the indicators addressed in this study: snow depth, the snow cover duration, the air temperature at 2 meters and total precipitation.

Anomalies are computed using the 1985-2015 climatology as reference (absolute differences for snow cover duration and temperature, relative differences for precipitation and snow depth). Taylor diagrams are calculated using anomalies for the common available time period : 1985-2015 (also used in Section 3.1).

For the trend analysis, we use a subset of in-situ snow observations stations with less than 10% missing values during the 1971-2019 snow period (taking the monthly values from November to April) (see Figure 1d and section 2.3.1 for more details).





For the same reason, we use the homogenized version of E-OBS (v19.0HOM) for air temperature at 2 meters. Linear trends are estimated using the Theil-Sen non-parametric estimator, and trends are considered significant if their 95% confidence interval excludes zero.

### 535    3.3.1    Representation of the interannual variability in the evaluated dataset

Figures 11, 12, 13 and 14 show the time series of winter (November to April) anomalies for temperature, precipitation, snow depth and snow cover duration. Solid lines with circle markers present the geographical mean value of the annual anomalies over each subregion, and the shaded areas corresponds to the standard deviation of anomalies within the subregion. For each subregions and datasets, Taylor diagrams based on winter anomalies for each subregion for the 1985-2015 period are also
displayed.



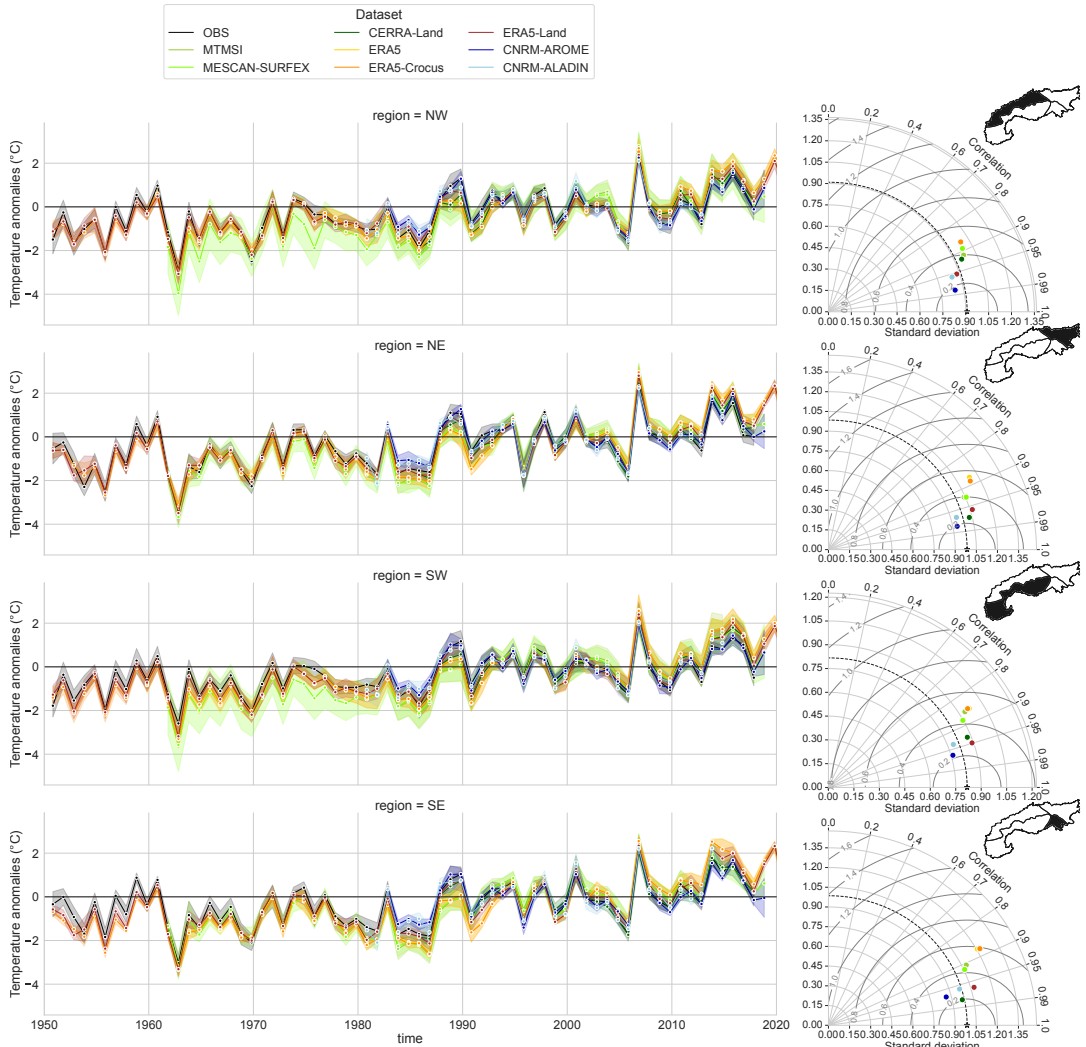

**Figure 11.** Anomalies of mean winter (November to April) values of temperature in the European Alps for the whole available time period for each dataset compared to the mean winter values of the 1985-2015 period. Continuous lines show the time series of the spatial mean of annual anomalies for a given subregion, shaded areas represent the standard deviation of annual anomalies for all grid points in a region. On the right, Taylor diagram calculated using the anomalies of winter values of temperature for the 1985-2015 period. The reference used is described in section 2.3.1.




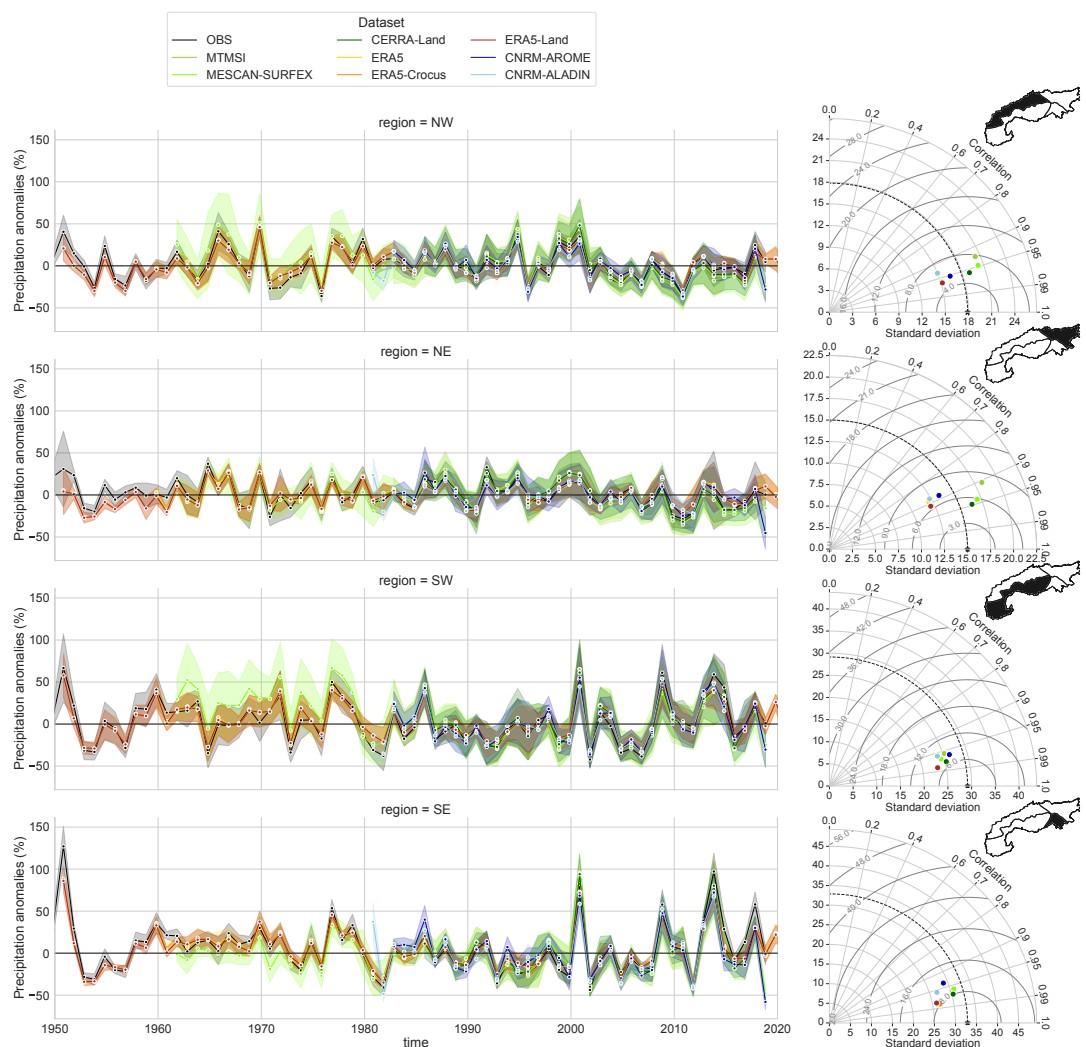

**Figure 12.** Same as Figure 11 but for precipitation.



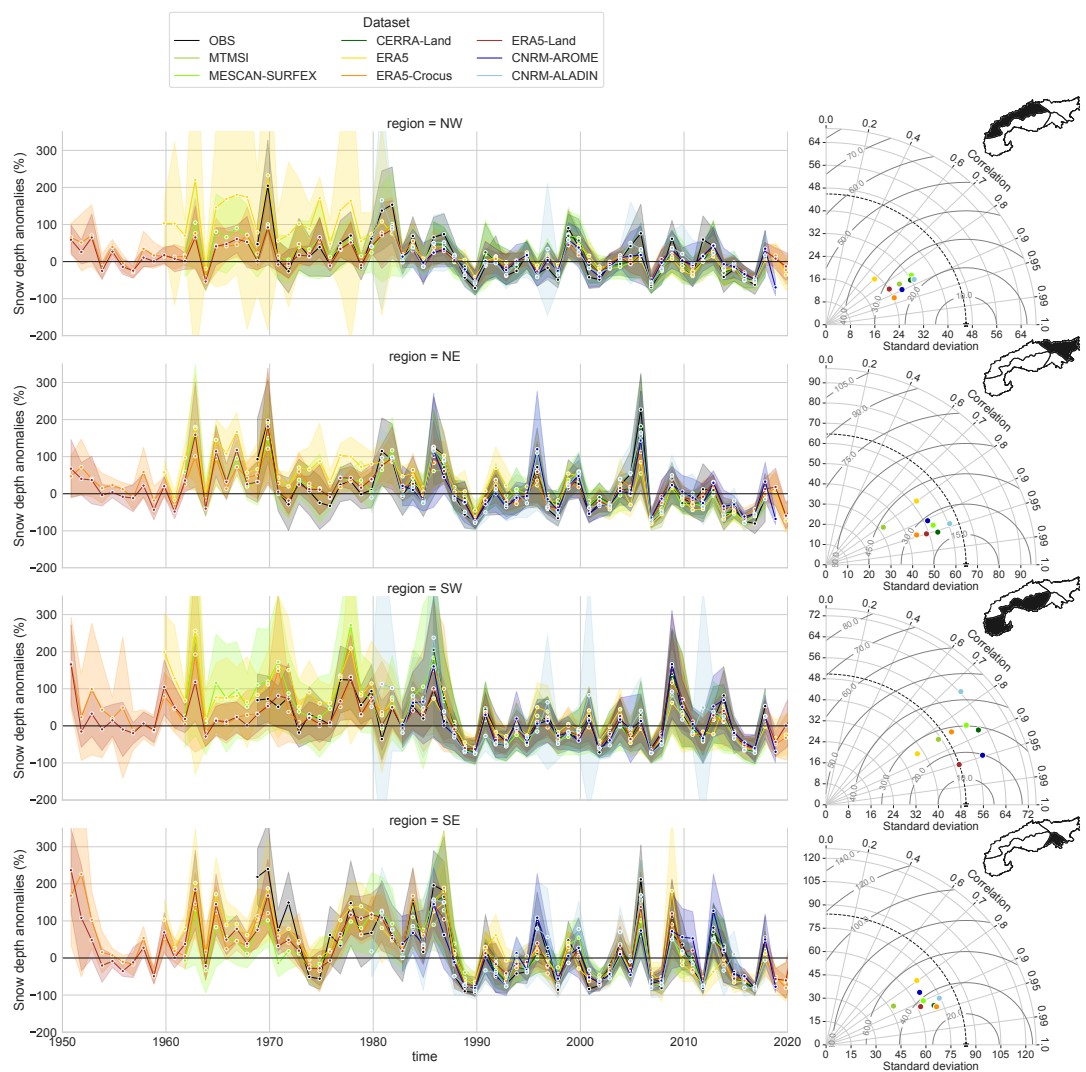

**Figure 13.** Same as Figure 11 but for snow depth.



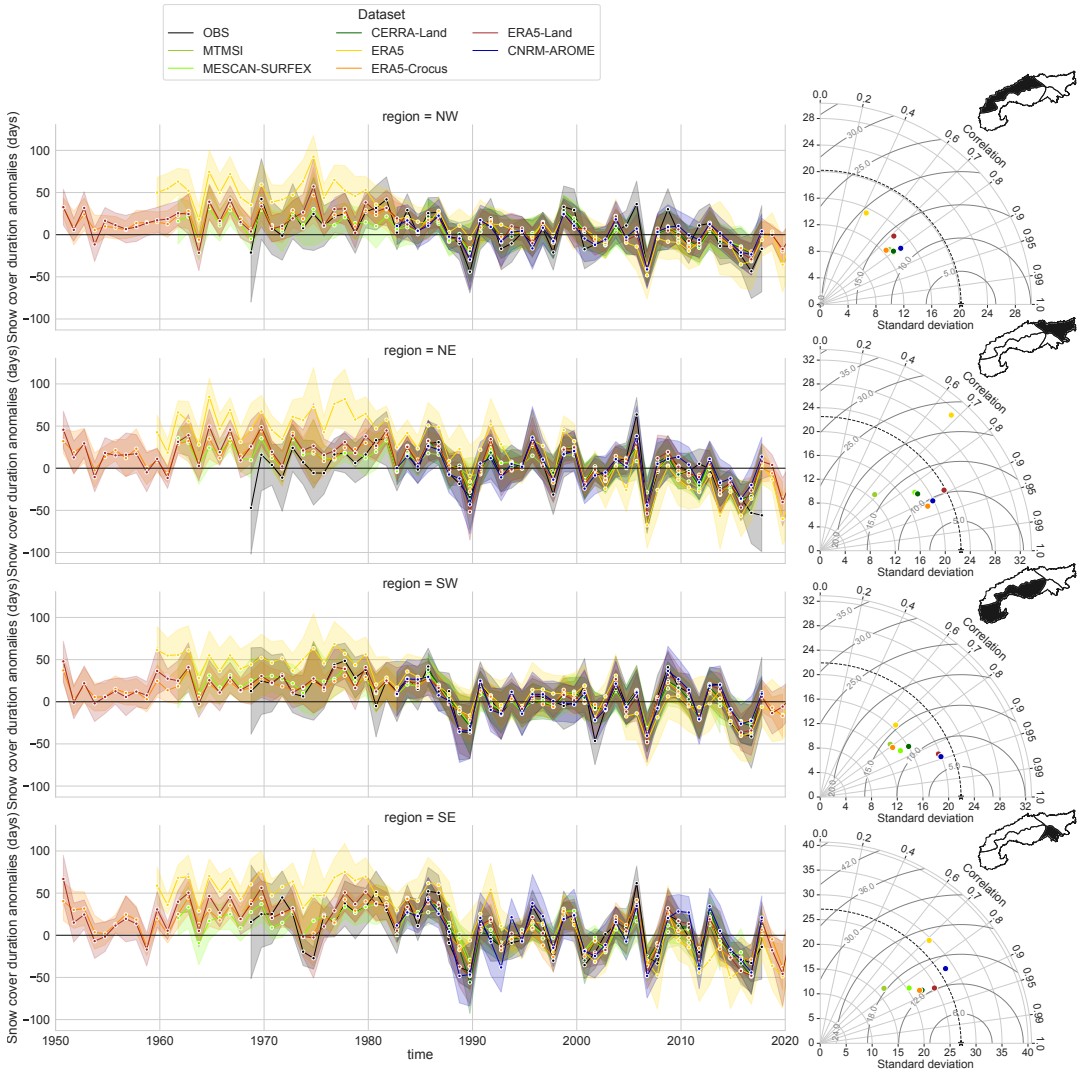

**Figure 14.** Same as Figure 11 but for snow cover duration (SCD).

Temperature anomalies of all subregions on Figure 11 show high correlation values (always above 0.8), with standard deviation of anomalies close to the standard deviation of the reference (0.8 to 1.0 ± 0.2 °C, i.e. less than 20% of difference with the reference) meaning that anomalies are of comparable amplitudes between datasets. The standard deviation displayed in shaded areas surrounding the solid lines on the graph, represents the spatial variability of anomalies inside a given sub-

region. On Figure 11 it is rather small (± 0.2°C at most) compared to the magnitude of the anomalies, and the anomalies follow comparable fluctuations among regions, indicating that most of the interannual variability occurs at a large spatial scale, affecting similarly the different subregions, and displaying a rather small variability inside a given subregion. Climate simulations (CNRM-AROME and CNRM-ALADIN) and ERA5-Land show the highest scores of correlations. CNRM-AROME and



CNRM-ALADIN slightly underestimate the amplitude of the anomalies, while ERA5-Land is closer to the reference in this
respect, or with a slight overestimation. ERA5, MESCAN-SURFEX, MTMSI and CERRA datasets show the lowest correla-
tions values, and generally overestimate the amplitude of anomalies. In a comparison of temperature anomalies errors between
multiple datasets including MESCAN-SURFEX, ERA5, E-OBS (v19.0 HOM) and COSMO-REA6 (a regional reanalyses at
6 km of horizontal resolution that does not assimilate near-surface temperature) against a set of homogenized in situ observa-
tions over the Swiss Alps, Scherrer (2020) came to similar conclusions : the lowest scores (higher mean errors in his case) are
found in winter and at elevations above 1000 m for MESCAN-SURFEX and ERA5, whereas COSMO-REA6 does not present
significant error differences between summer/winter and low/high elevation. The explanations put forward are twofold : (i)
the insufficient resolution of ERA5 that does not allow to capture the altitudinal gradient of temperature anomalies that can be
strong in winter where there are interplays between synoptic flow and local topography, and (ii) the assimilation of near-surface
temperature for MESCAN-SURFEX that seems to be problematic in high elevation regions. We note that in the eastern part
of the Alps (region NE and SE), CERRA provides better scores than MESCAN-SURFEX, and it may be the result of a more
selective assimilation procedures in this region (i.e. the assimilation of fewer and higher quality observations).

Precipitation anomalies (Figure 12) show a good agreement between datasets with all correlation values above 0.90 (ex-
cept CNRM-ALADIN and ERA5 for the NE subregion with a value of 0.85) with a relatively small intra-subregion variability,
ranging from 5 to 30%. However, there are strong differences between subregions. A graphical comparison of the time series of
anomalies (Figure 12) shows that none of them follow the same fluctuations, implying that interannual variability of precipita-
tion have a strong subregional component within the European Alps. Regarding the standard deviation of anomalies, as well as
the spread of the datasets on the Taylor diagram, a North-South partitioning can be made. Rather small values of the standard
deviation (10% to 20%) are found in the Northern part. There, MESCAN-SURFEX, CERRA and MTMSI datasets overes-
timate the amplitude of anomalies; CNRM-AROME, CNRM-ALADIN and ERA5-derived indicators underestimate it. The
southern subregion shows standard deviation of 25 to 33% with a generalized underestimation of the amplitude of anomalies
compared to LAPrec.

Snow depth anomalies (Figure 13) show lowest agreements between datasets than temperature and precipitation anomalies,
with correlation values varying depending on the subregion considered. Correlation scores range from 0.60 to 0.80 for the
northern subregions, 0.75 to 0.95 for the southern subregion, with an amplitude of anomalies underestimated by all the datasets
compared to the in-situ observation dataset for all subregions excepted the southwestern subregion. At every scale (subregion
and intra-subregion), snow depth exhibits a higher variability than temperature or precipitation. The amplitudes of anomalies
of mean winter snow depth spans ± 100%, with peaks at ± 300%. The spatial variability within a given subregion is large,
varying from 20% to almost 100% of the amplitudes of the anomalies, meaning that compared to temperature and precipitation,
a significant part of the interannual variability operates at a local scale. Comparing the amplitude and the correlation of the
anomalies with the reference dataset does not allow to identify a dataset, which would outperform all others. However, ERA5
systematically exhibit the lowest correlation values, and the strongest underestimation of the amplitude of anomalies in the
western part of the Alps. We also note that between 1950 and 1980, ERA5 show signicantly higher anomalie values along with
a large increase of their standard deviations specifically for the western region (northwestern and southwestern), a behaviour





specific to ERA5. It is due to some low elevation grid points (below 1000 m), showing a strong decrease of their mean winter
snow depth values between before 1980 and afterwards, that may be for a part an artifact induced by the strong increase of the
number of assimilated snow depth observations after the 1980's.

Snow cover duration anomalies (Figure 14) present the lowest agreement among the datasets. Correlation values are com-
prised between 0.4 for the lowest, 0.9 for the highest, mostly ranging between 0.7 and 0.8. Similarly to the snow depth anoma-
lies, the spatial variability of the anomalies are large at an intra-subregional scale, with a standard deviation of anomalies of
the same order of magnitude as the anomalies themselves. The amplitude of anomalies are also underestimated by most of
the datasets for most of the subregions as shown by the ratio of the standard deviation between the datasets and the reference
within the Taylor diagram Figure 14. No dataset outperforms the others in terms of correlation scores, RMSE and ratio of
the standard deviation, just as for snow depth, but ERA5 systematically shows the lowest scores. Correlation scores of ERA5
barely exceed 0.7, and drops to 0.4 for the northwestern region, indicating a low agreement concerning the oscillations of the
anomalies. Additionnaly, ERA5 snow cover duration anomalies also shows the strong shift of the mean values of anomalies
between the before 1980 and afterwards, that we relate to the same causes as for the snow depth.

### 3.3.2    Running window trend analysis

This section compares the long-term trends in the snow variables (snow depth and snow cover duration), temperature and
precipitation. Trends are calculated using the mean monthly values for the winter period (November to April) using the Theil-
Sen linear regression, and significance levels assessed using the boundaries of the confidence interval at 95% : if it excludes
zero, the trend is considered statistically significant.

As for the section 3.2, CNRM-ALADIN is not included for the SCD indicators because daily fields of snow depth values
are not available.

Figure 15 and 16 display the trend values for the winter values of temperature, precipitation, snow depth and snow cover
duration for each dataset, calculated using different window time durations (10, 20, 30, 40, 50 and 60 years), with a moving
central year of the window, by steps of 5 years. The longest time period, with a duration of 60 years, is obtained for ERA5,
ERA5-Land and ERA5-Crocus, spanning 1961-2020, centered on 1990. For all variables, trends computed with a 10 years
window length show large values, with strong fluctuations from one time period to the next (shifted by 5 years). The increase
of the window length leads to an increasing number of trends detected as significant, a decrease in the trend values and a
progressive loss of these fluctuations. This is related to the progressive smoothing of the effect of the shorter scale interannual
variability, increasing the ratio signal over noise and thus, letting longer scale variability and forced response of the climate
system becoming dominant in the value of the trends.

Winter temperature trends (Figure 15) calculated on window length of at least 30 years show a temperature increase over
the Alps, varying in their amplitudes from 0.1°C/dec to 1.0°C/dec depending on the years and datasets considered with a
generalized increasing warming over the recent decades. The trends values are highest for ERA5 and ERA5-Crocus dataset
regardless of the window length or the central year of the window used for calculation, with 30-years long trends varying
between 0.1°C/dec centered on 1970 ($\pm$ 15 years) to 1.0°C/dec centered on 2005 ($\pm$ 15 years) and 60 years long trends between



0.4 and 0.6°C/dec. Trends obtained from MESCAN-SURFEX, CERRA and MTMSI are close to ERA5 and ERA5-Crocus and rather similar trend values are found for climate simulations CNRM-AROME and CNRM-ALADIN, and ERA5-Land for
window length between 10 to 30 years. In a study comparing interannual variability and trends of temperature over Switzerland from multiple datasets against a set of homogenized in-situ observations, Scherrer (2020) shows that the homogenized version of E-OBS provides the closest estimates of anomalies and trends for the Swiss Alps, with winter trend values around 0.3°C/dec. Pepin et al. (2022) reviewed mountain temperature and precipitation trends worldwide and found values for the temperature trends vaaying from 0.2 to 0.5°C/dec over the last decades for the European Alps. Winter temperature trends calculated here
with E-OBS are in line with these findings, with values close to 0.3°C/dec on a 50 to 60 years long window length. This implies that the other datasets (MESCAN-SURFEX, ERA5, ERA5-Land, ERA5-Crocus) tend to sligthly overestimate this trend.

     Winter precipitation trends (Figure 15) display small values with only few trend values, compared to temperature trends, computed as significant, considering a window length of 20 to 60 years long. Calculated on a 10 years long time periods, trends can exhibit reversed signs from two consecutive time periods with values varying from -30 to +30%/dec. Even trends
calculated on a 30 years long window length shows periods of slight increases and slight decreases thereafter, with trend values lower than ± 10%/dec. We note that MESCAN-SURFEX, CERRA and MTMSI provide opposite trend values compared to the other datasets regarding the sign of the trend for a 10, 20 and 30 years time period centered on the year 2005 (2000 to 2010 for the 20 years time period). This behaviour may be due to the impact of the assimilation, through the heterogeneity in the number and quality of the assimilated observation. Nevertheless, it makes little sense to compute trend values over such short
time periods. Concerning the other datasets, they are in broad agreement concerning the signs and amplitudes of trends, in line with LAPrec taken as reference whatever the window length considered. Overall, long term (i.e. on a 40 to 60 years window length) winter precipitation trends are small with changes between +2%/dec and -7%/dec at most. Very few of these trend values are detected as significant to conclude to a change in the winter precipitation over the whole alpine ridge. Long-term subregional trends (see Appendix B Figure B2) show more contrasted results. Whereas the northern part only show weak and
alternating sign trends values always lower than 3%/dec, the southern part exhibits a winter drying trend over the last decades, with trend values ranging from -6 to -15%/dec, consistent among the different datasets. This decrease of winter precipitation in the southern part of the European Alps is in line with previous studies, such as Masson and Frei (2016), who calculated trends with EURO4M-APGD, a gridded precipitation dataset used to construct LAPrec dataset, and Ménégoz et al. (2020) who used regional climate simulations driven by ERA-20C (Poli et al., 2016).

Figure 16 displays a generalized decrease of the mean winter (November to April) snow depth over the recent decades ranging from -2%/dec to -25%/dec for window length of 40 years and above, showing higher discrepancies between datasets than temperature and precipitation trends. Similarly to the time series of anomalies (see Figure 13), fluctuations for trends in consecutive smaller time periods (10, 20 and 30 years) exhibit larger values, with frequent reversed sign of the trend values. In such cases, trend values for different datasets strongly diverge, with high differences concerning the trend values and even
cases of opposed signs. On window length higher than 30 years and for the common available years, MESCAN-SURFEX and ERA5 snow depth datasets show stronger trend values than the reference in-situ observation dataset, ranging from -9%/dec to -23%/dec compared to the -9%/dec to -13%/dec. For longer window lengths, ERA5-Land, ERA5-Crocus show trend values





closer to the trend values calculated using the observational dataset, from -5%/dec to -12%/dec. It is interesting that despite similar atmospherical forcings and a broad agreement in trend values for temperature and precipitation, there are such large

differences between the three ERA5-based datasets. Overall, the snow depth negative trend is already widespread over window lengths of 30 years and generalized for all datasets on window lengths from 40 to 60 years. For 40 to 60 years window length, trend values range from -2%/dec to -25%/dec, but mostly around -7%/dec to -15%/dec. These values are consistent with the recent literature on this topic using "purely" observational datasets. Indeed, Fontrodona Bach et al. (2018) found mean winter snow depth to have decrease from -5%/dec to -25%/dec over the northern part of the European Alps between 1951-2017 using

the ECA&D observational dataset. Matiu et al. (2021), who gathered the observational dataset used as snow depth reference in this study, have also computed trends, and found them to vary on average between -6%/dec and -10%/dec over the 1971-2019 period for the European Alps stations below 2000 m elevation.

Snow cover duration trends are in line with the mean winter snow depth trends for long-term window length (40 years and above), with a generalized disdecline over the last decades ranging from -3 days/dec to -24 days/dec depending on the year and dataset considered.

Nevertheless, if short-term trends calculated on a 10 years window length exhibits a similar behaviours as snow depth, with large values and divergence between consecutive periods and between datasets for a given period, the decreasing duration of the snow season is already widespread since the 1980's on window length of 20 and 30 years. For longer window lengths (40 to 60 years), all datasets excepted ERA5 are in broad agreement with the reference, showing a decline mostly comprised between -5 days/dec and -10 days/dec, whereas ERA5 strongly overestimate it, with arange of values more

than twice the values of the other datasets for the same period. This strong overestimation has to be linked to the one also shown concerning the mean winter snow depth, and may be for a part induced by heterogeneities concerning the assimilation, strongly reducing the amount and duration of snow (closer to the observation) for the most recent decades. The long-term (last 50 years) decline around -7 days/dec provided by ERA5-Land, ERA5-Crocus, MESCAN-SURFEX and the reference dataset is largely in line with the literature on that topic. Indeed, this rate lies within the likely range of decline reviewed in Hock et al.

(2019) of -0 days/dec to -10 days/dec, also close to the -8.9 days/dec over the 1970-2015 period found by Klein et al. (2016) over 11 stations in the Swiss Alps, the average -7 days/dec simulated over Austria for the 1960-2019 period by Olefs et al. (2020), as well as the range of -4.5days/dec to -7.0 days/dec find by Matiu et al. (2021) in the 1971-2019 period using a large set of in-situ observations over the whole Alps.







**Figure 15.** Winter trend values for the whole European Alps calculated using Theil-Sen linear regression for 2 variables (Temperature and Precipitation). The horizontal axis shows the central year used for the computation of the trend values. The vertical axis provides the series of datasets used for various lengths of the window time period (10, 20, 30, 40, 50 and 60 years) used for the computation of the trend values. A black framed square indicates a trend detected as statistically significant (confidence interval at 95% excludes zero). Precipitation trends are expressed in relative (%/dec) decrease or increase compared to the mean of the period, while temperature trends are provided as warming rate (°C/dec).





**Figure 16.** Winter trend values for the whole European Alps calculated using Theil-Sen linear regression for 2 variables (Snow depth and Snow cover duration). The horizontal axis shows the central year used for the computation of the trend values. The vertical axis provides the series of datasets used for various lengths of the window time period (10, 20, 30, 40, 50 and 60 years) used for the computation of the trend values. A black framed square indicates a trend detected as statistically significant. Snow depth trends are expressed in relative (%/dec) decrease or increase compared to the mean of the period, while snow cover duration trends are provided in days per decade (days/dec).

### 3.3.3 Elevation dependant climate change (EDCC) and spatial variability of the trends over the 1968-2017 period

Following analyses are dedicated to the spatial variability of the trends : their elevational gradients and their spatial distribution. Trends are calculated over the 1968-2017 period using the available datasets for this time period : OBS (i.e. in situ observations of snow depth for the mean winter snow depth and snow cover duration, E-OBS v19.0-HOM for the winter temperature and LAPrec for the winter precipitation), ERA5, ERA5-Land, ERA5-Crocus and MESCAN-SURFEX. The choice of a window length of 50 years rely on the analyses presented in section 3.3.2, showing that at least 40 years is the minimum time duration

required to get a consistent signal at the scale of the whole Alps.



Winter temperature trends on Figure 17 and 18 show different spatial patterns depending on the dataset. E-OBS HOM, here taken as the reference displays a rather smooth winter temperature trend field on Figure 18 with only few trends detected as significant (i.e. Non-hatched areas). Not surprisingly, the three ERA5 products show similar patterns with higher warming rates ranging from 0.5°C/dec to 1.0°C/dec in the eastern half of the Alps, all detected significant compared to the lower trends (i.e.

below 0.4°C/dec) elsewhere. MESCAN-SURFEX exhibits the highest warming rates almost everywhere, with stronger values over the Alpine ridge, corresponding to higher elevation areas, which can also be noticed on the boxplot Figure 17 through the strong elevational gradient. Giving the amplitude of the elevational gradient simulated by MESCAN-SURFEX above 1500 m (i.e. median shifted by 0.6°C/dec/1000 m), and the fact that the other datasets agree more on a slightly decreasing elevational gradient at these elevations similar to what Rottler et al. (2019) found over the Swiss Alps for the 1981-2017 period, MESCAN-

SURFEX behaviours may be considered as spurious artifacts.

Winter precipitation trends are weak for all datasets excepted MESCAN-SURFEX. For the reference and the three ERA5 products, spatial pattern are similar on Figure 18, with a widespread light decrease of precipitation, higher in the Western part of the Alps but still lower than 7% (not detected significant) and no elevational gradient on Figure 17. MESCAN-SURFEX trend field Figure18 shows larger values, exceding ± 15%/dec with a patchy appearance (i.e. alternating strong positive/negative

values) and a strong elevational gradient, both more reminiscent of artifacts than a realistic pattern.

Snow variables (i.e. snow cover duration and mean winter snow depth) present similar geographical patterns (see Figure 18). ERA5-Land and ERA5-Crocus expose a rather homogenous decline of both variables, around -10%/dec and -10 days/dec respectively for the mean winter snow depth and the snow cover duration, values that are close to the one given as reference using the set of in-situ observations. As shown above in section 3.3.2, ERA5 displays spurious values, almost twice the refer-

ence values, that we link to an heterogenous assimilation procedures that have corrected an overly thick modelled snowpack over the last decades, artificially reinforcing the decreasing trend. MESCAN-SURFEX snow variables trend fields show similar patterns as its precipitation trend field, with patchy field displaying a strong horizontal gradient, probably inherited from the precipitation field. Concerning the elevational gradient on Figure 17, all datasets are in broad agreement for both variables. They simulate a stronger relative declined of the mean winter snow depth at intermediate elevation, near the snowline. Snow

cover duration trends also present a weak elevational gradient for all the datasets, with a stronger decline at low and intermediate elevation, with a median near -10 days/dec compared to the -8 days/dec at high elevation, for ERA5-Land, ERA5-Crocus, MESCAN-SURFEX and the reference. The highest changes of snow conditions at low and intermediate elevations (near the snowline) have already been documented in multiple studies. Indeed, Kuhn and Olefs (2020) review of EDCC over the European Alps show that the declined of both snow cover duration and snow depth over the 1961-2018 period is linearly related

with elevations (i.e. with a stronger decline at low and intermediate elevations).



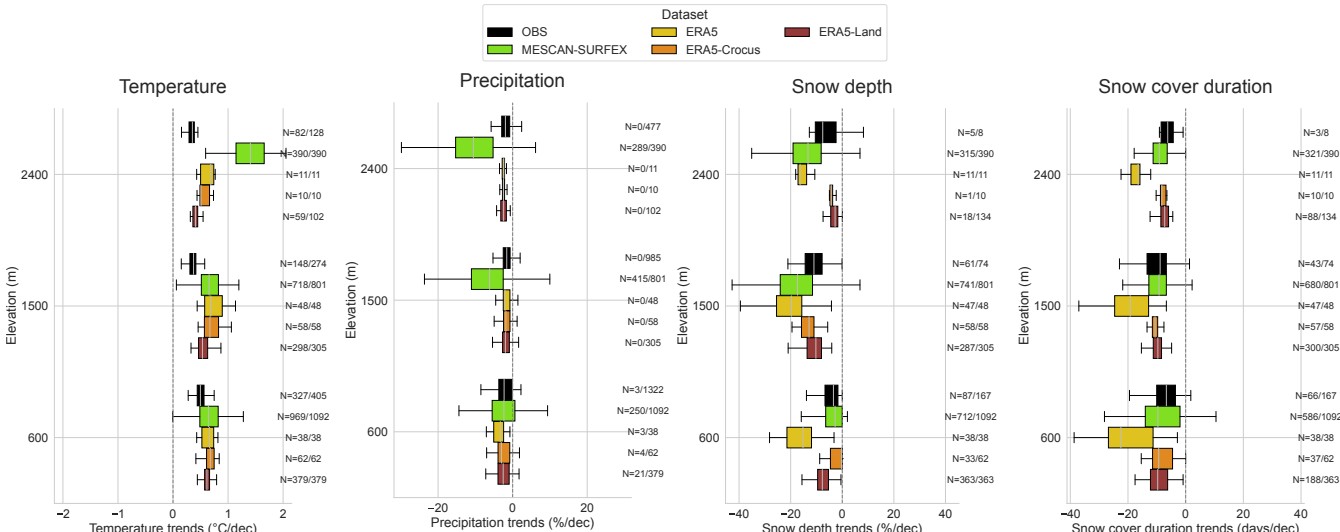

**Figure 17.** Boxplot of the winter trend values calculated over the 1968-2017 period (50 years) for each grid points of the whole Alps (grid points included within the 450 m - 2550 m elevation range) using Theil-Sen linear regression for 4 variables (temperature, precipitation, snow depth and snow cover duration). For this length of the period, five datasets are available : observations (E-OBS HOM v19.0 for temperature, LAPrec for precipitation and the set of in situ observations for snow depth and snow cover duration), MESCAN-SURFEX, ERA5, ERA5-Land and ERA5-Crocus. Precipitation and snow depth trends are expressed in relative (%/dec) decrease or increase compared to the mean of the period, while temperature and snow cover duration trends are provided as absolute rate.





**Figure 18.** Maps of the winter trend values calculated over the 1968-2017 period (50 years) for each grid points of the whole Alps (grid points included within the 450 m - 2550 m elevation range) using Theil-Sen linear regression for 4 variables (temperature, precipitation, snow depth and snow cover duration). Non-hatched areas indicate trends detected as statistically significant (the confidence interval at 95% excludes zero). Precipitation and snow depth trends are expressed in relative (%/dec) decrease or increase compared to the mean of the period, while temperature and snow cover duration trends are provided as warming rate (days/dec).

## 4 Discussion

In this study, we have compared the results of various modelling systems (global and regional reanalyses, and regional climate model simulations driven by a global reanalysis) against observations references (in-situ, kriged datasets and satellite observation). The comparisons are performed in terms of monthly and seasonal snow depth values, and snow cover duration, snow





onset date and snow melt-out date, addressing multi-annual averages of regional and sub-regional mean values, their inter-
annual variations, and trends over various time scales. Here we discuss the main findings of the study, and draw implications
for further use of these datasets in various contexts. We remind that this study investigates mean seasonal to monthly snow
characteristics and their main driving variables (temperature and precipitation) at the scale of the European Alps, and for four
sub-regions. The diversity of the datasets involved in this study and the difficulties to carry an unbiased point scale (station
vs. grid points) comparison over mountainous terrain (i.e. due to the scarcity of the observations and to the strong horizontal
gradients) conducted us to restrain the investigation to statistics over large areas. Consequently, the study does not address the
characteristics of the datasets at local scale, and these may differ from their large scale characteristics.

### 4.1  Snow cover characteristics as seen by different modelling systems

This subsection is dedicated to the discussion of the different behaviours of the datasets concerning the snow conditions,
through the height of snow and the seasonality of the snow cover (i.e. snow cover duration, snow onset date and snow melt-out
date).

Compared to the observational references, the worst performances (lowest scores and strongest deviations from the refer-
ence) concerning the representation of the snow characteristics are obtained with ERA5-Land, CNRM-AROME and CNRM-
ALADIN. CNRM-AROME and ERA5-Land both overestimate the snow depth, the snow season lasts too long (about 1 month
later than the other datasets) and with areas with spurious snow accumulation above 2000 m. For CNRM-AROME, these issues
have already been documented by Monteiro et al. (2022) over the French Alps ; the finding is found to also apply at the scale
of the European Alps. Monteiro et al. (2022) attributed these issues to the deficiencies of the snow scheme physics (i.e. missing
processes that lead to an underestimated melt), along with a biased surface energy balance related to both a misrepresentation of
surface-atmosphere exchanges and the underestimation of upwelling infrared radiation (due to the underestimation of nightime
cloud cover). Muñoz-Sabater et al. (2021) evaluated snow depth results of ERA5-Land and ERA5 against in-situ snow depth
observations, mainly in the Rocky mountains (Western US). They do not report any snow accumulation issues at high elevation,
and for low and intermediate elevations, ERA5-Land shows a lower mean absolute error compared to ERA5. In their study,
only 30 observations points were used in the European Alps, and show a higher mean absolute error in ERA5-Land results
compared to ERA5, in line with our analyses. This was interpreted as an effect of the dense SYNOP observations network in-
corporated in the snow assimilation system of ERA5, allowing it to perform better than ERA5-Land over this region. Compared
to CNRM-AROME, at a regional to sub-regional scale, no strong overestimation of precipitation was shown in ERA5-Land,
which could explain the spurious accumulation of snow, but they have in common the use of a single layer snow scheme, and
a strong temperature cold difference in winter. The simplicity of these snow schemes, through the underestimation of melting
processes, along with atmospherical forcing leading to too much snow accumulation (likely overestimation of winter/spring
precipitations) and weaker melting conditions (underestimation of winter/spring temperatures and erroneous surface mass bal-
ance) could explain part of the observed differences in snow depth values compared to the reference observations. Despite the
use of the same snow scheme, it is likely that these errors are strongly reduced in ERA5 thanks to the data assimilation of
snow depth observations. Orsolini et al. (2019) showed that over the Tibetan Plateau where no observations of snow depth are





assimilated, ERA5 strongly overestimates the snow cover and the snow depth, and attribute part of it to the overestimation of

snowfall. These gives us evidence that assimilation (snow depth but also precipitation and near-surface temperature) potentially strongly benefits to ERA5 results in the European Alps. In ERA5-Crocus, the use of a detailed representation of the snowpack within the Crocus snow model is intended to provide a better representation of the accumulation and melting processes. ERA5 and ERA5-Crocus, potentially for different reasons, provides snow depth values and simulate a seasonality of the snowpack closer to the reference snow cover than ERA5-Land, although with a coarser horizontal resolution.

The combination of a strong negative temperature bias in CNRM-ALADIN and its strong overestimation of winter precipitation (already identified by Terzago et al. (2017) and common to numerous RCM of EUROCORDEX) is a possible explanation for the overestimation of snow accumulation at the beginning of the season for intermediate and high elevation bands. The explanation for its too early melt is, however, rather due to issues with the snow scheme and would require further sensitivity studies to precisely identify the causes.

The regional reanalyses MESCAN-SURFEX, CERRA and MTMSI, provide snow depth values and snow seasonality closest to the reference. Although they do not assimilate snow depth values, it appears that their surface analysis through the MES-CAN system provides reliable snow cover first order atmospherical drivers (precipitation and temperature) along with a fair representation of the snow physics within ISBA-ES or Crocus.

Overall, our analysis finds that there are multiple combinations of atmospheric and land surface modelling systems (Essery
et al., 2020; Menard et al., 2020) and their corresponding data assimilation frameworks (or not), leading to various snow cover datasets generated through numerical modelling. The performance of these datasets vary, and none is totally found irrelevant for the variables analyzed in this study. However, no system is found to outperform all of the other ones in all of the dimensions of the evaluation, which can also be traced to the fact that none of the currently available systems combines the "best" of all worlds (high resolution, sufficiently sophisticated snow cover model well adapted to the land surface model, surface data
assimilation including precipitation, snow cover data assimilation etc.).

## 4.2   On the complexity of assessing differences between modeling system

In addition to the general discussion provided above, here we insist that the comparison between all of the evaluated datasets, and in particular the three ERA5 datasets and CNRM-AROME, is not straightforward. Even if ERA5 and ERA5-Land have in common their land surface model and snow scheme configuration, ERA5-Land does not simulate the feedback from the
surface to the atmosphere at its resolution, while ERA5 is constrained by a large number of observations through its land data assimilation system. ERA-Crocus is run in standalone mode too, but the configuration of the land surface snow scheme is somewhat specific (e.g. it considers fully snow covered surface even if only a thin layer (i.e. 10 cm) of snow is presents). CNRM-AROME shares a rather similar single layer snow scheme than ERA5-Land and ERA5, but simulates a coupling with near-surface atmosphere without assimilating any observations. Finally, HTESSEL (for ERA5 and ERA5-Land) and SURFEX
(for CNRM-AROME, ERA5-Crocus) are two different land surface models, based upon different ways of taking into account the presence of snow over the surface, particularly its impact on energy balance calculations. This implies a wide range of responses of the different snow schemes (notably according to their complexities) for atmospheric forcings that can be of





very variable quality as underlined by Terzago et al. (2020). Furthermore, even in the case of a similar simulation, different use of the same snow scheme can be done, leading to wide differences on the simulated snowpack (Napoly et al., 2020). 790 Thus, a clear attribution of the different factors conducive to the different behaviours of the modelling systems turns out to be virtually impossible in the absence of an extensive sensitivy study to apportion the contribution of each possible factors, which is beyond the scope of this study (and may remain an elusive goal due to the complexity involved in disentangling such complex modelling frameworks).

## 4.3    Representation of the interannual variability

The study of the interannual variability of the snow conditions through two variables, the mean winter snow depth and the annual snow cover duration as well as their main driving variables (near surface temperature and precipitation) shed light on multiple aspects of the variability of the snow conditions in the European Alps.

Near surface winter temperature anomalies are rather homogenous among the different subregions with a small intra subregion variability, meaning that most of the interannual variability affect similarly the whole Alpine region. All datasets reproduce 800 correctly the anomalies compared to the reference, and even if no strong discrepancies are found between them, climate simulations (CNRM-AROME and CNRM-ALADIN), CERRA-Land and ERA5-Land performs better than ERA5, MTMSI and MESCAN-SURFEX. Winter precipitations show a strong subregional variability, varying from $\pm$ 50% from one year to the next, but rather similarly inside a given subregion. This is coherent with past classifications such as HISTALP (Auer et al., 2007), making winter precipitation varying for a significant part independantly from one subregion to the others. On these 805 anomalies, datasets only show little discrepancies with LAPrec taken as reference, and no dataset outperforms the others in every subregions.

Mean winter snow depth and snow cover duration show the largest interannual variabilities, across and within subregions of the European Alps. Datasets show the lowest correlation scores with the reference value, with a generalized underestimation of the amplitude of the interannual variability. In fact, both indicators are influence by a large set of components (temperature, 810 precipitation, wind, radiation etc.) and result from the integration, over the course of the winter season, of many processes operating at multiple space and time scales. These contribute to widening the discrepancies concerning the snowpack characteristics between near location all along the winter periods and increase the interannual variability at a very local scale. Additionally, the accuracy of precipitation and temperature fields throughout the snow season as well as the snowpack modelling system are widely different across the datasets, and this ultimately determine a large fraction of the quality of the simulated snow depth 815 time series. Overall, these analyses demonstrate that the snowpack interannual variability is still challenging to simulate and as the interannual variability of precipitation and air temperature is already rather well represented according to our analysis, efforts might be directed towards the improvement of the snow modeling framework.

## 4.4    Reliability of the trend values

The analysis of mean winter values at the scale of the whole Alps assessed in section 3.3.2 confirmed that detecting a trend 820 needs to be done cautiously. First of all, our analysis confirms that trends analyzed over too short time periods (typically, less





than 30 years), are prone to large errors and the influence of decadal climate variability. This is illustrated on Figure 16, which confirms that, in particular, snow cover trends analyzed over the past 20 years only, cannot be considered representative of longer-term climate trends. As a consequence, That some key data records, such as satellite observations, only cover such a limited time span, should not be a reason for assessing trends over irrelevant time periods. Instead, our study shows that there
is potential to use satellite information together with other sources of information (in-situ, numerical simulation, and their combination), to overcome the time resolution/target variables conundrum (Gascoin et al., 2022).

Further than this reminder on some key safeguarding principles related to the calculations of snow cover trends, this analysis bears some relevance with respect to the sources of variability and trends for snow cover variables in mountain regions. Indeed, the different variables investigated are affected at different time scale (i.e. from annual to multi-decadal) by fluctuations, that
can alternatively favor particular climate conditions (e.g. warm and dry or cold and wet conditions). These oscillations of the climate system generally occurs at a larger scale than the European Alps, and can be describe through modes of variability. The two main modes that affect the variables investigated here on the Alpine ridge are the North Atlantic Oscillation (NAO) and the Atlantic multi-decadal oscillation (AMO or AMV) (Cassou et al., 2021). They typically favor during winter and in their positive phase, warmer and wetter conditions for the NAO, colder and wetter conditions for the AMO implying a
potentially strong influence on the snowpack evolution (Scherrer and Appenzeller, 2006; Durand et al., 2009). However, while a relationship between past decrease of spring precipitation in southern Alps and the positive phase of the AMO have been reported (Brugnara and Maugeri, 2019), as well as winter precipitation in the Western Italian Alps with the NAO (Terzago et al., 2013), to our knowledge, only a few studies have attempted to link alpine snow conditions to large-scale modes of variability (NAO, AMO, ENSO... ). Due to the low amplitudes of these oscillations and the large number of superimposed factors affecting
the snowpack at different spatio-temporal scales, no clear consensus has been found so far. Overall, our analyses show that below 30 to 40 years time length, much of the trends are not detected as significant and can fluctuate between strong negative or positive values for consecutive time periods. Beyond 40 years window lengths, the signal/noise ratio becomes larger and can lead to the detection of meaningful and significant changes in the mean values.

This study also brought to light artifacts affecting the trends held by datasets incorporating information from observations.
ERA5 (and to a lesser extent ERA5-Land) and MESCAN-SURFEX winter temperature trends are systematically overestimated compared to the reference E-OBS and past studies. Figures 17 and 18 show that in the case of ERA5 products, warming is particularly overestimated in the Eastern part of the Alps, whereas MESCAN-SURFEX simulated a much stronger warming at high elevations, leading to a spurious elevation dependant warming. Similarly, MESCAN-SURFEX winter precipitation trend fields display unrealistic geographical distribution of values associated with strongly overestimated amplitudes. These
spurious precipitation trends are reflecting in comparable snow depth and snow cover duration trend fields, also overestimated. Finally, ERA5 snow depth and particularly snow cover duration trends are strongly overestimated (twice the amplitudes of the reference).

These behaviors point at least in part to a common cause : the assimilation of a varying (in quantity and in quality) number of observations overtime, better correcting model biases in spaces and times where observations are dense. This leads to a shift
in the distribution of a variable, therefore inducing an artificial trends.



ERA5 and MESCAN-SURFEX are based on modelling strategies that favor the maximisation of the agreement between model and observations upon the conservation of a consistant timeline allowing the product to be used for trend assessment.

Users of these products should therefore be aware of the limits of the quality of a datasets incorporating information from observations. As reminds by Cornes et al. (2018); Kaiser-Weiss et al. (2019); Bandhauer et al. (2022), their qualities highly 860 vary depending on the spatio-temporal scale considered (i.e. their effective resolutions), and the areas considered, both strongly affected by the density of the observations used in the product. In addition, the use of an heterogenous number of observations can lead to a differential correction overtime of the products, inducing artificial trends within it, as we see here for the Alps with MESCAN-SURFEX winter precipitation trends, or ERA5 snow variables.

## 5 Conclusions

In this study, we investigate the ability of various modelling systems (global and regional reanalyses, and regional climate model simulations driven by a global reanalysis) to represent past snow conditions and its main atmospherical drivers over the European Alps, using observation references (in-situ, kriged datasets and satellite observation). The comparisons are performed in terms of monthly and seasonal values, addressing multi-annual averages of regional and sub-regional mean values, their inter-annual variations, and trends over various time scales.

The comparisons of the datasets over a common period of 30 years (1985-2015), in terms of the average of monthly and seasonal values of the snow depth and snow cover duration at a regional scale, shed light onto a variety of behaviours that can lead to strong differences between the evaluated and the reference datasets. CNRM-AROME and ERA5-Land are both found to overestimate the amount of snow and its persistence along the snow season, a phenomena that increase with elevations, leading to spurious snow accumulation above 2000 m. CNRM-ALADIN shows an overestimated accumulation of snow at the 875 beginning of the season and a total melt that occurs about one month earlier than the reference dataset for elevations above 1500 m. ERA5, ERA5-Crocus, CERRA-Land, UERRA-MESCAN-SURFEX and MTMSI simulate a monthly evolution of the snow rather close to the reference, confirms at a regional and subregional scale, with no dataset systematically outperforming the others. The spatial comparison against MODIS remote sensing data supports what appears to be a common issue of all datasets, namely the overestimation of the snow cover duration above 1000 m, with stronger errors near the snow line (around 880 1500 m) and concentrated on the melt-out date.

The discrepancies concerning the past climatology of snow conditions found between the reference and evaluated datasets may have multiple sources, affecting differently each of the datasets, and we can only hypothesize them here. Indeed, setting a clear attribution of the different factors leading to the different behaviours of the modelling systems would required multiple extensive sensitivy studies adressing independantly the impact of multiple error sources. Among them, we identified 885 biases from the atmospherical forcings, limitations of land surface data assimilation procedures for mountainous areas and the misrepresentation of surface-atmosphere exchange and key snow processes (i.e. wind-transport, glacier accumulation...) along with potentially inadequate configurations of the snow schemes leading to erroneous melt rate.





The study also addresses the representation of the inter-annual variability by the different evaluated datasets, and their simulated trends over various time scales. Temperature and precipitation interannual variability are found to be rather well represented by most of the datasets. However, the simulation of the interannual varibility of snow cover variables appears to remain difficult for all datasets. Their lowest performance may come from several causes, related to the difficulty of representing the strong variance of the inter-annual fluctuations, and the local variability of the snow conditions. This may be due for a part on misrepresented processes, related to the problems mentioned above, related to the specificity and the complexity of the modelling of the snowpack. We mention that snow variables are somehow specific, as they result from a cumulative history all along the snow season. Therefore, errors from atmospherical forcings, snow modelling and coupling between surface-atmosphere and assimilation also cumulate over time.

The computation of trends over various time scales leads us to reiterate some safeguarding principles on the matter. Indeed, computing trends over time scale shorter than 30 years result into the detection of noisy signals, strongly affected by inter-annual to multi-decadal variability, and often not statistically robust. In addition, some datasets (MESCAN-SURFEX and CERRA-Land for precipitation, snow variables and ERA5 for snow variables only) lead to spurious trends (i.e. strongly over-estimated amplitudes and incoherent geographical patterns even for long-term trends), probably induced by spatio-temporal heterogeneities in their assimilation procedures. These datasets should be used carefully for climate change applications as some of the simulated variables may be affected by artifacts, impacting the reliability of the resulting trends.

Overall, no dataset outperforms the other in terms of the multiple aspects investigated here. Upstream modelling choices have indeed great consequences on the downstream uses of these datasets. Reanalyses (and datasets based upon observation sources) hold the potential to partially fullfill the gap of in situ observations in mountainous areas and providing a consistent and reliable baseline of the key variables describing the evolution of climate conditions in mountainous areas. Nonetheless, we find that the quality of these datasets is scale- and location-dependent. As mentioned by Isotta et al. (2015), Kaiser-Weiss et al. (2019) and Bandhauer et al. (2022), the quality of the datasets incorporating observations (using assimilation procedures or direct kriged method) is strongly dependent upon the density of the observations used (and their spatio-temporal homogeneity), and the effective resolution of these dataset is in fact generally lower than the provided grid resolution. In mountainous regions the scarcity of observations remains a strong obstacle, however standard assimilation techniques can lead to deteriorating the quality of the simulated fields, as the spatial variability is high over mountainous areas, and the influence of an observations should be restrained to its domain of applicability (same elevation, slope, aspect etc.). Furthermore, observation-based prod-ucts (including reanalyses) necessarily result from a trade-off between reproducing a climatically relevant time evolution and usingthe maximum number of available observations to produce the best possible description of the state of the atmosphere and snow cover at any given time. Ultimately, this type of dataset continues to rely on the density and quality of the past obser-vational network, which cannot be extended retrospectively. Here we also evaluated the results of regional climate simulations driven by a robust larger-scale (global) reanalysis. Our results indicate that this can offer an alternative to computationnally intensive high-resolution reanalyses, for climate studies. Indeed, they can simulate climatically homogenous atmospherical and surface conditions (depending on the quality of the driving, larger scale reanalysis), and even provide an estimate of the modelling uncertainties if combined with a set of simulations from different regional climate models, although they do not



directly incorporate high resolution data through a data assimilation framework. However, our study shows that for mountain regions several issues with such models need to be tackled. In particular, be it for high resolution climate modeling or numerical

weather prediction models used for the production of reanalyses, investigations into the behaviour of the snow cover scheme (and their broader relationship with the rest of the land surface models interacting with the atmosphere) are crucially required. This will hopefully enable to address these issues for forthcoming versions of these modeling systems.

*Code and data availability.*   All computations were performed with Python software version 3.9.13. The codes are available from a repository (Github repository) which includes scripts (in a notebook form) for the following tasks : performing all data preprocessing, reading the

different data sources, statistical analyses and figures making.

For the snow depth in situ observations availability, please refer to this article : https://doi.org/10.5194/tc-15-1343-2021. LAPrec v1.1 dataset is available on the Copernicus Data Store following this doi : https://doi.org/10.24381/cds.6a6d1bc3. E-OBS v19.0 HOM and E-OBS v23.1 are available for download following this URL : https://surfobs.climate.copernicus.eu/dataaccess/access$_e obs_m onths.php$. Remote sensing MODIS (MOD10A1F) dataset is available following this doi : https://doi.org/10.5067/MODIS/MOD10A1F.061. CNRM-AROME

and CNRM-ALADIN hindcast simulations can be downloaded on the ESGF website (https://esgf-node.ipsl.upmc.fr/projects/esgf-ipsl/, accessed 1st february 2023). ERA5 dataset is available on the Copernicus Data Store following this doi : https://doi.org/10.24381/cds.adbb2d47. ERA5-Land dataset is available on the Copernicus Data Store following this doi :https://doi.org/10.24381/cds.e2161bac. MESCAN-SURFEX dataset is available on the Copernicus Data Store following this doi: https://doi.org/10.24381/cds.32b04ec5. The MTMSI dataset is available on the Copernicus Data Store following this doi: https://doi.org/10.24381/cds.1ac1b4ba. ERA5-Crocus can be accessed upon request to the

corresponding author.



# Appendix A: Reference characteristics of the Alpine snow cover

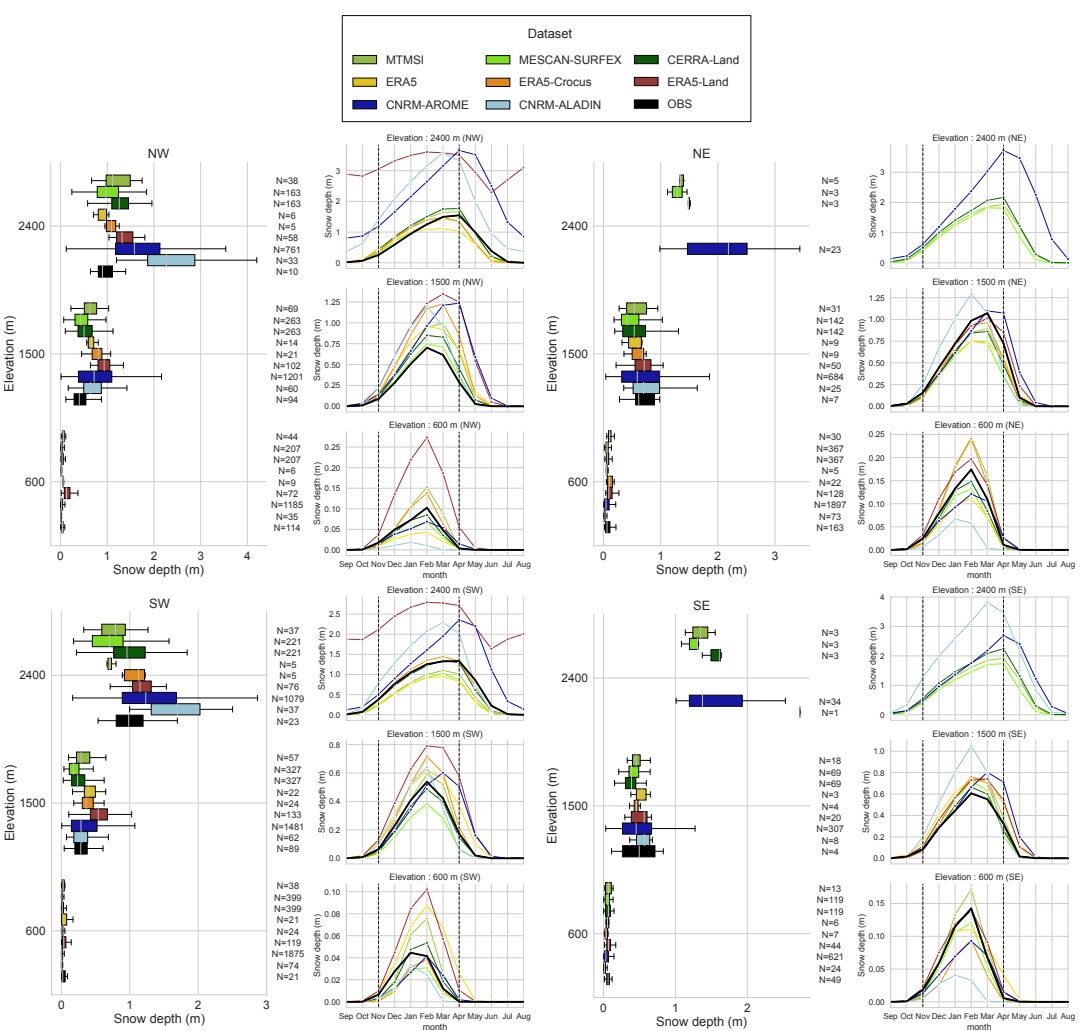

**Figure A1.** For each subregions of the European Alps : boxplot representing the spatial distribution of mean winter season (November to April) snow depth values over the 1985-2015 period for each dataset for three elevation bands (600 m ±150 m, 1500 m ±150 m, 2400 m ±150 m). Annual cycle of the mean monthly snow depth values for the 1985-2015 period for each dataset, for three elevation bands (600 m ±150 m, 1500 m ±150 m, 2400 m ±150 m).



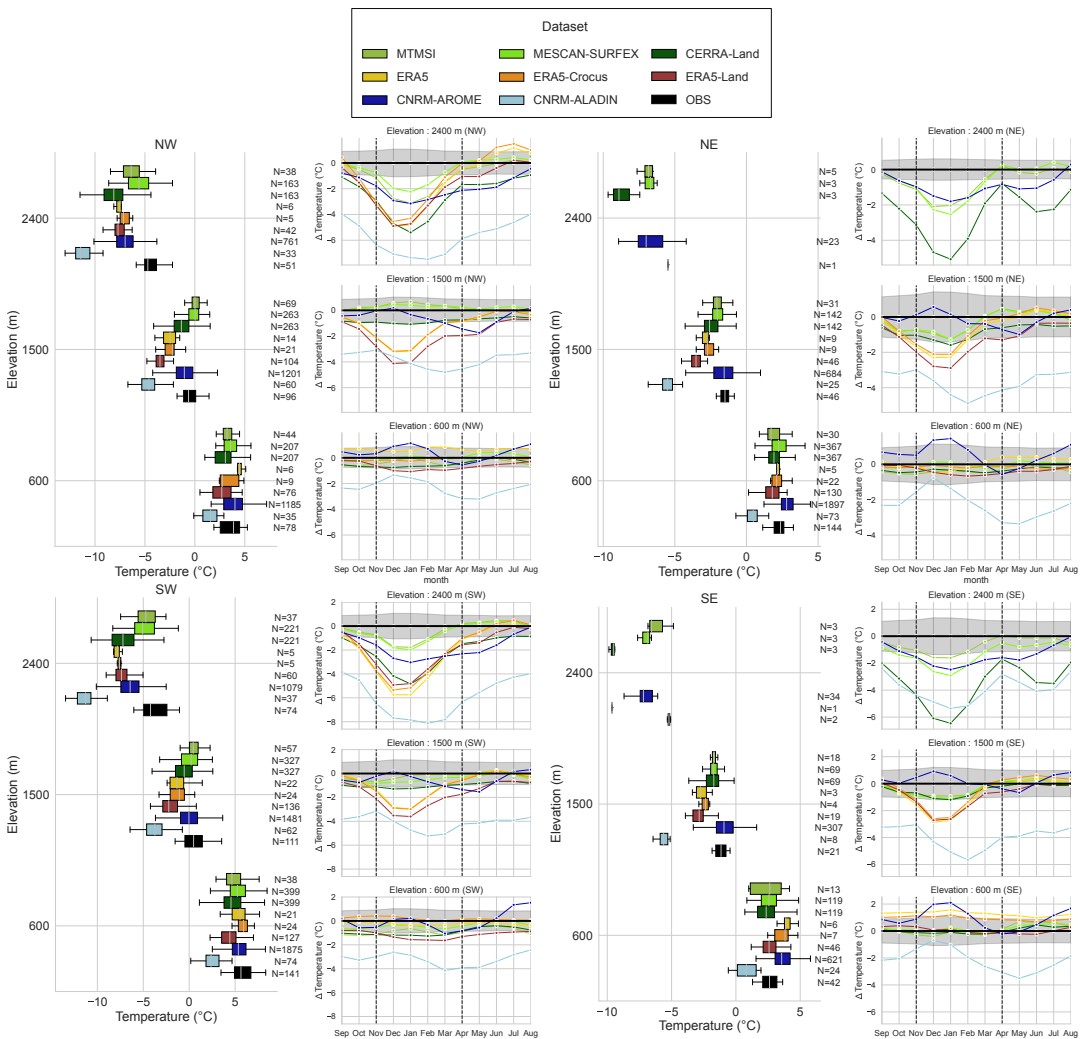

**Figure A2.** For each subregions of the European Alps : boxplot representing the spatial distribution of mean winter season (November to April) temperature values over the 1985-2015 period for each dataset for three elevation bands (600 m ±150 m, 1500 m ±150 m, 2400 m ±150 m). Annual cycle of the mean monthly error values of temperature for the 1985-2015 period for each dataset, for three elevation bands (600 m ±150 m, 1500 m ±150 m, 2400 m ±150 m).



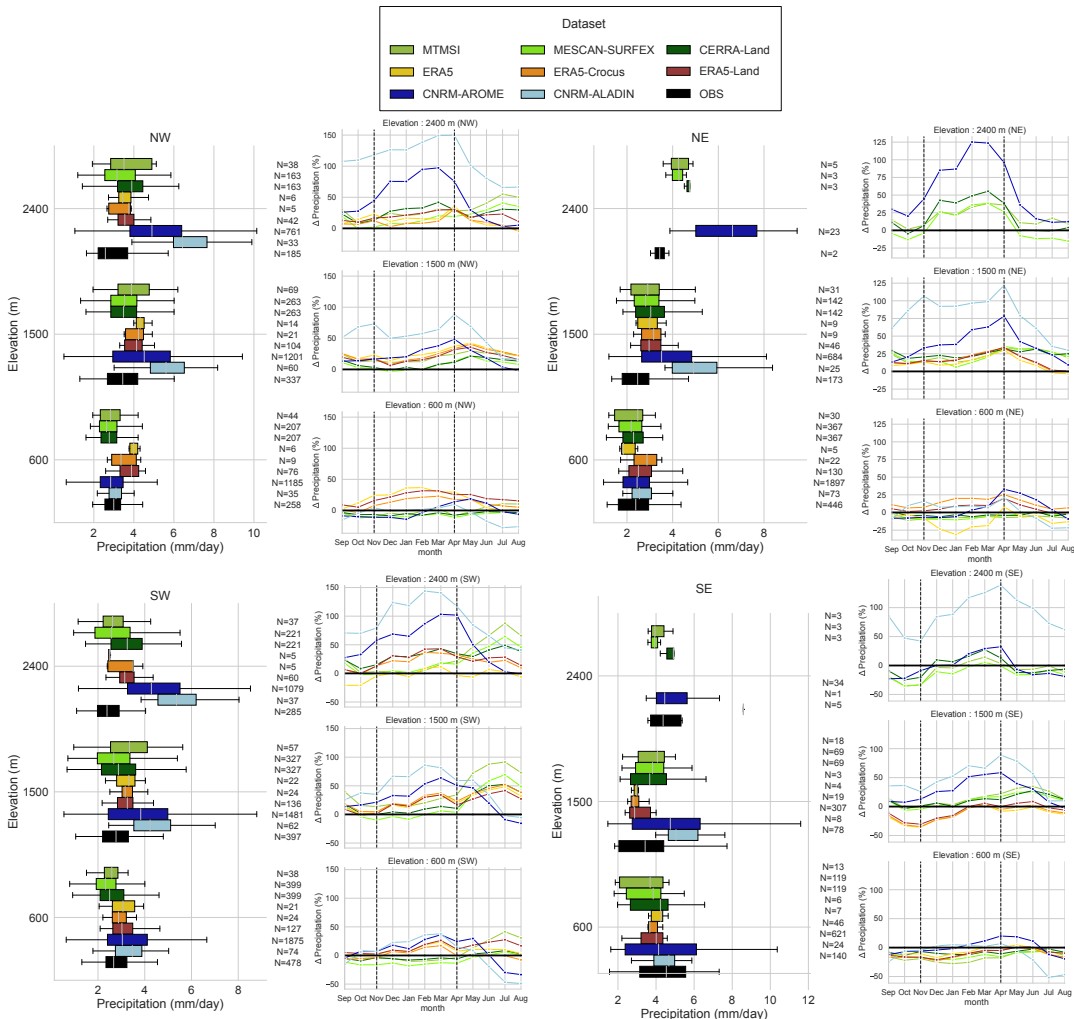

**Figure A3.** For each subregions of the European Alps : boxplot representing the spatial distribution of mean winter season (November to April) precipitation values over the 1985-2015 period for each dataset for three elevation bands (600 m ±150 m, 1500 m ±150 m, 2400 m ±150 m). Annual cycle of the mean monthly error values of precipitation for the 1985-2015 period for each dataset, for three elevation bands (600 m ±150 m, 1500 m ±150 m, 2400 m ±150 m).



## Appendix B: Interannual variability and trends



**Figure B1.** Winter trend values for each subregions of the European Alps calculated using Theil-Sen linear regression for the air temperature at 2 m. The horizontal axis shows the central year used for the computation of the trend values. The vertical axis provides the series of datasets used for various lengths of the window time period (10, 20, 30, 40, 50 and 60 years) used for the computation of the trend values. A black framed square indicates a trend detected as statistically significant. Trends are provided as warming rate per decade (°C/dec).




**Figure B2.** Winter trend values for each subregions of the European Alps calculated using Theil-Sen linear regression for the precipitation. The horizontal axis shows the central year used for the computation of the trend values. The vertical axis provides the series of datasets used for various lengths of the window time period (10, 20, 30, 40, 50 and 60 years) used for the computation of the trend values. A black framed square indicates a trend detected as statistically significant. Trends are provided in relative rate per decade (%/dec).



**Figure B3.** Winter trend values for each subregions of the European Alps calculated using Theil-Sen linear regression for the snow depth. The horizontal axis shows the central year used for the computation of the trend values. The vertical axis provides the series of datasets used for various lengths of the window time period (10, 20, 30, 40, 50 and 60 years) used for the computation of the trend values. A black framed square indicates a trend detected as statistically significant. Trends are provided in relative rate per decade (%/dec).



**Figure B4.** Winter trend values for each subregions of the European Alps calculated using Theil-Sen linear regression for the snow cover duration. The horizontal axis shows the central year used for the computation of the trend values. The vertical axis provides the series of datasets used for various lengths of the window time period (10, 20, 30, 40, 50 and 60 years) used for the computation of the trend values. A black framed square indicates a trend detected as statistically significant. Trends are provided in days per decade (days/dec).



*Author contributions.* The study was defined by SM and DM. The formal analysis was performed by DM with input for methodology from SM. The original draft was written by DM and SM.

*Competing interests.* The authors declare that they have no conflict of interest.

*Acknowledgements.* The authors gratefully acknowledge the WCRP-CORDEX-FPS on Convective phenomena at high resolution over Europe and the Mediterranean [FPSCONV-ALP-3]. This work is part of the Med-CORDEX initiative (http://www.medcordex.eu). CNRM/CEN is a member of LabEX OSUG@2020.

We acknowledge the E-OBS dataset from the EU-FP6 project UERRA (https://www.uerra.eu, last access: 23 January 2023) and the 950 Copernicus Climate Change Service and the data providers for the ECA&D project (https://www.ecad.eu, last access: 23 January 2023).

We thanks the following institution for sharing with us daily snow depth records from station data : the Piemonte Dipartimento Rischi Naturali, the Ufficio neve e valanghe della Regione Autonoma Valle d'Aosta, the Società Meteorologica Italiana (www.nimbus.it), the meteorological office of the Slovenian Environmental Agency, the ARPAV- Avalanche Centre of Arabba- Italy and the ARPA FVG – OSMER eGRN.

We thanks Cécile Caillaud for data curation of CNRM-AROME and many supports on its exploitation, and Simon Gascoin for the enlightening discussions and advice on the use of MODIS satellite data.





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
