# Peer review of "Multi-decadal evaluation and analysis of past winter temperature, precipitation and snow cover information over the European Alps from reanalyses and climate models"

_EGUsphere, 2023_

## Author Comment (AC1)

**Responses to RC1 on manuscript EGUSPHERE-2023-166**

*Authors :*
Diego Monteiro, Samuel Morin

**1 Reviewer comments 1**

Monteiro and Morin provide an in-depth analysis of snow cover and its main drivers temperature and precipitation in regional and global reanalyses and two climate models over the European Alps. They use in-situ observations and MODIS for snow cover, and state-of-the art spatial data sets for temperature and precipitation. Its the most extensive evaluation over the European Alps, both in terms of models and reference data, and highly relevant to understand the potential and limitations of reanalyses (and of the two climate models) for modeling and observing snow cover information in complex terrain. The authors produced a wealth of information and the manuscript is well written with great figures.

The manuscript is of high relevance and technically sound. I have only a few general remarks, which are related to the authors use of diverse data sets at different spatial resolutions and associated issues in complex Alpine terrain. Looking forward to seeing the revised version and the authors answers. (Im sorry for the length of the review, but the authors paper is also quite extensive.)

**

We thank reviewer 1 for detailed and insightful comments and suggestions, which have lead to improving the quality of our manuscript. Below we provide point-by-point answers to the suggestions and comments.

**1.1 General comments**

**1.1.1 Choice of elev bands : did you carry out a sensitivity analysis ? E.g., there is the break between elev mismatch from 600m to 900m, partly from 1500 to 1800 (Fig 3).**

As mentioned in the section 2.4.2 of the original submission, two arbitrary choices have been made for the analysis of the results :

1. The width of the elevation bands is set at 300 m. This results from a trade-off between the elevation range within a a given elevation band for a given region, and the number of grid points or observations within a given elevation band. Setting a wider width would lead to the combination of grid points or observations spanning a too large elevation range (particularly for snow depth and temperature) and make our analyses coarser. Setting a lower width would lead to too few observations or grid points per elevation band. Note that we use the same width here that the SAFRAN/Crocus (S2M) reanalysis developed for the French mountainous regions and for which this elevation width was found a good compromise.

2. The choice of 3 elevation bands used for the graphical representation of our results makes it possible to display results for different environments in the Alps : 600 m for the valleys and low elevation hills, 1500 m for the intermediate elevation near the snow line and 2400 m for the high mountain conditions. The very ephemeral snow cover below 450 m has lead us to set 600 m as our lower level elevation band. For the high elevation conditions, we were restricted by the coarse resolutions of some of the evaluated (i.e. 30 km for ERA5), including too few points above 2500 m.

In order to work with equally-spaced elevation bands and maximize the number of in situ observations of snow depth, we chose to focus our analyses over the 600 m, 1500 m and 2400 m elevation bands. However, note that we use grid points or observation stations within +/- 150 m above and below each elevation band (i.e., the 600 m elevation band brings together observations and grid points spanning from 450 to 750 m elevation). The 900 m and 1800 m elevation bands could have also been relevant to qualify the low elevation hills and valleys and the intermediate elevation respectively, but including too many elevation bands in the main manuscript would be detrimental to its conciseness, hence the need to choose a few relevant elevation bands.

However we introduce here further analysis spanning more elevation bands. We calculated scores (ME, MAE and Correlation) similar to the Figure 6 of the manuscript, for the three variables investigated, for all elevation bands ranging from 600 m to 2400 m. This is shown on Figure 1.

[Figure]

FIGURE 1 – Mean values and scores of mean errors (ME), mean absolute errors (MAE) and correlations calculated using mean monthly values from November to April over the 1985-2015 period for the whole Alps and compared to the reference (in-situ observations dataset for the snow depth, E-OBS for the temperature and LAPrec for the precipitation) for each dataset for multiple elevation bands by steps of 300 m elevation (600 m $\pm150$ m to 2400 m $\pm150$ m). Scores are presented for snow depth, temperature and precipitation, and scores of MAE and biases are normalized by the mean values of the reference (given in %) for snow depth and precipitation

For the three variables, differences with reference values (i.e. error scores, ME and MAE) increase with elevation, common to all datasets. Winter temperature are lower than the reference above 1200 m, but ERA5 and CNRM-AROME show a slight warm bias at 600 m, that can not be seen in ERA5 at 900 m. Winter precipitation are larger above 1500 m, except for MESCAN-SURFEX, CERRA-Land and MTMSI below 1200 m. Winter snow depth is higher than the reference in much of the datasets, except below 900 m, where an underestimation can be seen for most of the datasets (excepted ERA5-Land).

For the three variables, the choice of the elevation band displayed (i.e. 600 vs. 900 m, or 1600 vs. 1800 m) would not change the main results of the manuscript, because the results are very similar in these cases. Largest differences are identified at the lowest elevations (between 600m and 900m), where models show specific behaviour (i.e. underestimation of precipitation for UERRA-based products, underestimation of snow depth for most of the datasets, and higher temperature values for ERA5 and CNRM-AROME), which are better illustrated at 600 m than 900 m elevation.

Following this complementary analysis, we keep the same elevation bands for the analysis and add in the revised manuscript that the results for intermediate elevation bands are consistent with the main patterns observed across the visualized bands.

**1.1.2 "Is there any influence of topographic smoothing in models in your findings, or did you circumvent this by only taking elevation averages ?"**

One of the reasons behind our analysis focusing on elevation bands is to mitigate some of the influence of topographic smoothing, such as horizontal shifts in meteorological patterns of low resolution model outputs compared to higher resolution reference products. Based on this approach, we can analyze, within spatially large areas covering a fraction of the European Alps, or the European Alps as a whole, elevation-dependent patterns and features of the datasets irrespective of the horizontal resolution - although topographic smoothing can not be avoided.

**1.1.3 All products are evaluated at their native resolution, right ? Does the horizontal resolution have an influence on elevational patterns ? Is there a dependency between horizontal and vertical variability ?**

Part of the answer to this question is in fact address in the previous reply. We keep all products at their native resolution because the scope of the study was to compare a wide variety of modelling strategies in their ability to recover past snow conditions over the Alps. In order to compare the different products we performed an elevation band analyses, as described above.

We understand the meaning of elevational patterns as the variability of mean values (or trends) between elevation bands (i.e. elevational gradients) for a given region (or for a subregion).

It is highly likely that horizontal resolution has an influence on the elevational patterns of atmospheric and snow variables. However, horizontal resolution only is not sufficient to explain the large variability of elevational patterns between our datasets. Among the factors that influenced the vertical variability we can cite :

1. Assimilation of observations.
2. Atmospheric dynamics in the models.
3. Model physics and parameterizations.

In this study, we can only illustrate and analyze differences between various configurations, but we cannot apportion the contribution of horizontal resolution only on elevational patterns. This would require to have a model run in the same configuration with only a varying resolution, which is not available to us and is thus beyond the scope of the present study.

From this study, we can not draw conclusions on a ńsystematic or generalized ż effect of the horizontal resolution on elevational patterns, and similarly (for the same reasons) for the dependency between horizontal and vertical variability.

**1.1.4 (optional) To reduce the length of the paper, the authors could also move some elements from the main manuscript into a supplementary (and also from the appendix to a supplement). A personally biased suggestion is : Figs 11-14, moving window analysis.**

We have taken this suggestion into account and move Figures 11 to 14 to appendices, and replace them by the anomalies and Taylor diagram for each variables, aggregated over the whole European Alps.

**1.1.5 Using spatial variability is problematic in the authors case, because of the different horizontal resolutions. Usually, with coarser spatial resolution, the spatial variability (and any measure such as the standard deviation) decreases, so apparent differences between spatial variability are likely artifacts from using different resolutions. Also Im skeptical about comparing spatial variability between sparse points and grids without first checking this issue. This also relates to the the use of Taylor diagrams with the reference being station data, which Im not sure is technically sound.**

We are aware that increasing resolution will most likely inevitably generate higher spatial variability. The purpose of the comparison focusing on spatial variability (Figure 10, section 2) is to corroborate that coarser grid size leads to undersampling the variability of the snow conditions within a given region. This is most likely related, to a great extent, to the horizontal resolution but can also come from several other factors as those described above (assimilations, dynamics, physics). For better clarity, we rephrased part of the text l.503 of the discussion manuscript : "From Figure 10abc, the size of the errorbars provides information on the spatial variability of the SOD and the SMOD. On this basis, we can conclude that it is underestimated in all datasets, excepted CNRM-AROME, but it may be partly related to the too coarse horizontal resolution in most of the datasets."

Also, we note that the Figures 7 and 8 show that increasing horizontal resolution does not necessarily lead to an increase of the spatial variability. In these figures, CNRM-ALADIN precipitation and snow depth values exhibit a similar (or even higher) horizontal and vertical variability than CERRA-Land, while the model resolution is almost more than twice finer. An other example is provided by the standard deviation values of the snow cover duration (SCD) in Figure 10 ; it is rather similar for CERRA-Land

and ERA5 whereas there is a factor 6 between their horizontal resolutions. Hence, horizontal resolution of the products alone does not necessarily drive differences in spatial variability of the snow cover.

We are also aware that taking sparse observations as a reference to assess spatial variability within a given region is not optimal and can only be underestimated. Nevertheless, it is the only reference we have access to, and we do not consider it as an absolute reference. The aim of this comparison is to assess the difference with the variability of the reference dataset. In the case where the variability of the tested dataset is lower than the reference, then it indicates that it is underestimated. Otherwise, we can not draw specific conclusions about it.

Figure 10 in Section 2 is the only direct comparison of the spatial variability of the snow cover duration between the datasets and the reference. In this figure, we also use MODIS SCD, which shows that spatial variability in the set of in-situ observations is also underestimated.

The Taylor diagrams are comparing the time series of the tested datasets, compared to a reference observation, of winter anomalies (at the scale of subregions or at the scale of whole Alps) over the 1985-2015 period. On this graph, standard deviation values represent the temporal variability of anomalies over this period. The main assumption here is that the set of in situ observations can be considered representative of the temporal variability of meteorological and snow variables.

Nonetheless, this comment has prompted us into improving the comparison of the aggregated datasets taking into account data for various elevation bands, for Figures 11 to 14 and for Figures 15 and 16. In the submitted manuscripts, we performed a simple average across elevation bands, without taking into account the hypsometry. This inevitably leads to placing disproportionate emphasis on elevation bands covering a small fraction of the entire domain. To circumvent this issue, we have revised our analysis and computed aggregated averages using weighting factors related to the hypsometry (based on a 100 m resolution DEM).

Accordingly, Figure 11 (in the revised manuscript) and Figures 15 and 16 have been updated, and the description of the methodology has been changed L.286 of the discussion manuscript : "Section 3.3 includes figures averaged over multiple elevation bands, at the scale of the whole Alps. To circumvent biases induced by the large differences in the representation of the hypsometry in the different datasets (Figure 3b), the averages over multiple elevation bands have been weighted by the relative fraction of each elevation bandusing the DEM at 100 m resolution as a common reference. This method ensures the same representativeness of each elevation band in the average value, regardless of the horizontal resolution of the dataset under consideration."

**1.1.6 At times, discussion and results are mixed. Also some methods are mentioned the first time in results. Note that a short repetition might still be useful, given the many different periods and methods used.**

In the revised version of the manuscript, thanks to this and other comments from the two reviewers, we believe we have prevented as far as possible the mixing of method and results, as well as results and discussion.

**1.2 Minor comments**

**1.2.1 (optional) I found the title rather uninformative given the huge amount of content of the paper. I guess its hard to synthesize such a broad assessment, but the authors might think of adjusting the title to reflect their study better (possible keywords that come to mind are reanalyses, (climate) model, observations, elevation, evaluation, ...)**

Accordingly, We modified part of the title to make it more informative : "Multi-decadal evaluation and analysis of past winter temperature, precipitation and snow cover information over the European Alps from reanalyses and climate model."

**1.2.2 General : The authors use the ellipsis extensively, especially in parentheses, but this is bad practice, since it leaves readers wondering what could mean. The authors should consider rewriting these in some way. A suggestion (not necessarily the best) could be using other sentence structures, such as L41 : climate variables, such as temperature, precip, SCA, and SWE, are needed**

The text was revised for better clarity.

**1.2.3 L6 : Attention, kriging a special form of spatial interpolation and EOBS and LAPrec have not (solely) used kriging. Better use another wording for observation based spatial data sets.**

We agree with reviewer 1 comment and replaced kriged dataset L.6 by "gridded observational dataset".

**1.2.4 L38-40 : wide range of changes..., some changes : vague, please clarify**

We reformulated the sentence L. 38-40 to "While many physical changes [...]"

For the second part "some changes", it may be unclear for readers so we rephrased L.38-40 into : "some physical processes remain imperfectly characterized such as the elevation dependant climate changes (Pepin et al., 2015, 2022), as well as numerous impacts on a large variety of related domains, such as water resources, ecosystems, natural hazards".

**1.2.5 L94 : Please check duplicated sentence parts.**

We thank reviewer 1 for his/her remark and corrected the manuscript accordingly.

**1.2.6 L105 : Check missing sentence.**

We thank reviewer 1 for his/her remark and corrected the manuscript accordingly L. 105 : "[...] and find that the quality of the dataset for a given area is for a large part determined by the number of assimilated observations."

**1.2.7 Fig1 : high elev points in b-d are hard to see. Maybe remove the white component of high elev.**

We removed the white component of the colormap.

**1.2.8 Sec 2.2. is more related to remote sensing, not variables of interest**

In this section, we mentioned all the variables and indicators that will be used in the study. Indeed, remote sensing takes a large part of the paragraph, related to the fact that the used indicators (SCD, SOD and SMOD) are derived from the NDSI that requires a specific pre-treatment, compared to the other variables (temperature, precipitation and snow depth), directly available from the products.

**1.2.9 L152ff unclear which years and which months were used in which analysis. Also missing values were allowed in the first case ?**

Concerning the first part of the analyses (section 3.1), we keep all stations data that have at least 70% daily valid values from December to April for the 1985-2015 period. This is the least restrictive filtering that we performed and omitted to mention it in the submitted manuscript.

The paragraph have been reworded for better clarity : "Section 3.1 focuses on the reference characteristics of the snow cover. In order to have the largest spatial coverage to compute monthly to seasonal mean snow depth over large regions, we keep all stations data that have at least 70% valid daily values over the November to April for the 1985-2015 period (see Figure 1b). Section 2 focuses on the snow cover seasonality, using the indicators SCD, SOD and SMOD computed using continuous daily values of snow depth over the winter period, and compare it to satellite observations from MODIS (record starting in 2000). Most of the missing values happen in summer (for most of the observation stations, no snow is present during this period). In this section, we keep stations with more than 80% of valid daily values over all year of the 2000-2015 period (see Figure 1c). Section 3 focuses on the interannual variability and trends. This requires the most homogeneous possible dataset along with a sufficiently long time period,

so we only keep stations with more than 90% valid daily values over the winter months (November to April) of the 1968-2018 period (see Figure 1d)."

**1.2.10    L206 and Fig2 : CHTESSEL vs HTESSEL for ERA5 ?**

We thanks reviewer for remarking this inconsistency. Figure 2 was corrected accordingly.

**1.2.11    Fig 3a : What is UE (Alps) ?**

UE Alps stands for European Alps, but it is unclear, so we replaced it by : "Whole Alps".

**1.2.12    Sec 2.4.3 : How did you define SOD, SMOD, and SCD ? Did you take a full hydrological year ? The same period as in the in-situ obs ? Did you deal with intermittent snow episodes at start and end, and if yes how ?**

SOD, SMOD and SCD are defined in section 2.2, and this applies to the rest of the study. The snow cover duration is defined as the longest consecutive period in a full hydrological year with a snow depth (or NDSI in the case of MODIS data) value above a given threshold. It means that intermittent snow episodes at the beginning or at the end of the season shorter than the longest snow-covered consecutive period are not considered. Note that in section 2.4.3, skill scores (TPR, TNR, FPR, FNR, Kappa and accuracy) are calculated using daily values over the entire year.

**1.2.13    Snow depth threshold : The problem with your choice of skill scores is that it is heavily influenced by the large amount of values with absence of snow cover (caused by low elev and I assume, because its not clear from your methods using the full year). This is why accuracy basically is the same as TNR. You might want to consider using PPV and NPV (positive, negative predictive value), since they adjust for prevalence (or alternatively limit the evaluation to less months). You can also consider dropping FPR and FNR, since they are equivalent to TPR and TNR (1 - *). Otherwise I would not put too much effort in the snow depth threshold, since its likely the least influencing factor in your study.**

Concerning the time period, we use every daily values available from April 1st of 2000 to December 31st of 2015. As suggested, we removed FPR and FNR and computed and added the PPV and NPV to the Figure 4a.

**1.2.14    L319ff, and fig 4b : unclear what difference you are looking at (between elev classes ? between MODIS and in-situ ?) and what the standard deviation refers to.**

Fig 4b show the error metrics (ME and MAE) of the SCD, SMOD and SOD between the set of in-situ observations (as defined in Figure 1.c) and the corresponding MODIS grid points for the 15 seasons (from 2000-2001 to 2014-2015). On this graph, snow cover duration from the in-situ observations is calculated with a varying threshold of snow depth to determine the absence or presence of snow (from 1 cm to 15 cm) while MODIS SCD is defined using a constant threshold of NDSI set at 20%. Solid lines represent the mean error metrics of all locations and seasons, and shaded areas around them, the standard deviations of these error metrics between elevation bands when gathered by elevations band of 300 m, from 600 m $\pm$ 150 m to 2400 m $\pm$ 150 m. The aim of displaying it was to see if error metrics vary with elevation, and if not, confirm the validity of the threshold for every elevations.

We recognize that this part may be unclear for readers so we rephrased L.319-320 into : "Figure 4b show the mean error metrics of the SCD, SMOD and SOD between the set of in-situ observations and the corresponding MODIS pixels for the 15 seasons (from 2000-2001 to 2014-2015). On this graph, snow cover duration from the in-situ observations is calculated with a varying threshold of snow depth to determine the absence or presence of snow (from 1 cm to 15 cm) while MODIS SCD is defined using a constant threshold of NDSI set at 20%. Solid lines represent the mean error metrics of all locations and seasons, and shaded areas around them, the standard deviation of these error metrics between elevation classes when gathered by elevations band of 300 m, from 600 m $\pm$ 150 m to 2400 m $\pm$ 150 m." Legends of Figure 4b was also updated for better clarity.

**1.2.15   L346 : AR1 is used if you have lagged effects (carryovers between years), which I assume to be negligible for annual values and the studied variables. So I do not see how it should help with heteroscedasticity ? Also unclear which trend estimates you show in the end ?**

According to Ribes et al., (2016), AR1 is a widely used assumption in climate sciences to account for persistence in climate data, where internal variability is considered to be a first-order autoregressive process. When compared with standard Ordinary Least Squares (OLS) or Theil-Sen (TS) estimator, it can provides information on how OLS or TS underestimate the values of the Confidence Interval (CI), and ultimately the number of trend values detected as significant. We used it in that sense in our study and found only little differences concerning the width of the CIs and the number of trends detected as significant, so we decided to use the Theil-Sen estimator for simplicity and reproducibility (as it is widely used in the climate science community).

We rephrased L.342-350 to improved clarity : "Section 3.3 compares trends for the different variables of interest using seasonal mean winter values (November to April) for the reference and the evaluated datasets. Trends are calculated using the robust nonparametric Theil-Sen estimator, insensitive to the changing variance of the residuals (Sen, 1968), along with a MannKendall test for significance assessment based on a $p$ value threshold at 0.05. In order to verify the robustness of the method, we tested the use of the standard ordinary least squares regressions (OLS), and the generalized least squares (GLS) with an autoregressive component (AR(1)) to account for the effect of the interannual variability on the calculated trends, known to lead to an increase of the size of the confidence intervals (Ribes et al., 2016), and thus, affect the number of trends detected as significant. The resulting trends were of comparable values for the three methods, but the OLS method leads to the detection of more significant trends, compared to the GLS with AR(1) and the Theil-Sen methods which generally share similar thresholds significance levels for our analysed time series."

**1.2.16   L356 : I could not find any supplementary material only appendix.**

We thanks reviewer 1 for the remark, this was a mistake, subregional figures were in appendix.

**1.2.17   L450 : Not sure. Fig A3 is very similar to Fig 8ghi, and inconclusive summer bias is only for low elev  at mid and high, its still a positive bias overall models.**

We agree that the summer bias is particularly inconclusive for low elevations, but the overestimation compared to the reference is generalized in western regions only, and therefore, does not appear as a systematic bias at mid and high elevations.

**1.2.18   L453ff : Is undercatch also a problem in summer at mid to high elev ? Im thinking strong winds and convective events ? Otherwise, how do you explain that summer biases at mid to high elev are in the same range or even higher than in winter ?**

To our knowledge, no systematic underestimation of liquid precipitation in the Alps related to the undercatch of rain gauges have been reported so far in theliterature, compared to the undercatch of solid precipitation, more frequent in winter. Nonetheless, we can not exclude the hypothesis. Part of the discrepancies in western regions may also come from the overestimation of convective precipitation in models that use a parameterization for deep convection events (i.e. all products excepted CNRM-AROME). Multiple clues point in that direction ; the reduced/absence of discrepancies between CNRM-AROME and the reference compared to the other products, and other studies (Isotta et al., 2015 ; Bandhauer et al., 2022) that reported an overestimation of summer precipitation in ERA5 and HIRLAM system (i.e. used to produce MESCAN-SURFEX, CERRA-Land and MTMSI datasets) compared to the same reference in the Alpine region.

Following the two comments on summer precipitation, we rephrased L.450-453 to : "Indeed, excepted CNRM-AROME, most of the datasets show an overestimation of summer precipitation in the western part of the Alps at mid and high elevations, while low elevations and eastern part of the Alps only show low and inconclusive differences. Note that the overestimation of summer precipitation for HIRLAM model (used to produce MESCAN-SURFEX, CERRA-Land and MTMSI), ERA5 and CNRM-ALADIN have already been reported in the literature (Isotta et al., 2015 ; Bandhauer et al., 2022 ; Monteiro et al., 2022), and related to their coarser horizontal resolutions and the parameterization of deep convection."

**1.2.19   Fig9 : Why do the numbers of grid cells not agree with previous figures ? Most striking is ERA5 and ERA5-Crocus at 600m, but also the others are different.**

This was due to an error in the code computing the number of grid cells. Figure 9 has been updated accordingly.

**1.2.20   L502 : Spatial variability difference is related to different resolutions. If you aggregate MODIS to the other resolutions, the spatial variability should decrease.**

This comment related to the major comments on the manuscript, addressd at the beginning of this response.

**1.2.21   L507 : Might be related to different station densities across the domain. Did you do a regional analysis also for this ? Would be interesting.**

We thank reviewer 1 for his comment. In order to see whether the mismatch in the low elevation snow season between MODIS and the set of in situ observations can be explained by oversampling of one subregion in the in situ observations, we have plotted on figure 2 the duration of snow cover at 600 m, for each dataset, for all sub-regions.

FIGURE 2 – Barplot whose edges represent the spatial mean over the Alps of the snow onset date (SOD) and the snow melt-out date (SMOD) for 15 seasons (2000-2001 to 2014-2015) for the 600 m ±150 m elevation band, for the four subregions, for each dataset at their native resolution. The errorbars surrounding the edges of each bar represent the standard deviation of the spatial distribution of mean values of the SOD and the SMOD.

On figure 2, all in situ observations show a seasonal lag in snow cover compared with MODIS for all subregions, although the shift is greater in southern regions than in northern ones. Thus, we cannot

conclude from this that the heterogeneous density of stations across the domain is the cause of the shift at low elevation.

**1.2.22 L528 : So the Taylor diagrams are based on the 1985-2015 period, or on the full ? If only for 1985-2015, wouldnt it make more sense to put them in one of the sections before ? (please also take into account the major comment above related to spatial variability)**

The Taylor diagrams provide 3 statistical indicators (i.e. the Pearson correlation, standard deviations and RMSE) calculated on pairs of time series of the same length. Figures 11-14 use the winter anomaly time series of the 4 variables investigated, and are designed to compare the representation of interannual variability between different products and reference datasets. The period 1985-2015 was chosen because it is the longest period common to all the products evaluated. Since these figures deal with the representation of interannual variability, we decided to keep them in this section of the manuscript.

Nonetheless, the purpose of these figures may be unclear for readers, so we added a sentence L.540 to make it clearer : "These figures are designed to compare the representation of the interannual variability between the different products and the reference data sets."

**1.2.23 Fig. 11-14 : Very hard to read since strong overlaps. Also it is unclear why here you put regional plots instead of an elevational analysis like before (to me, there are little differences between regions in these figures)**

The aim of plotting it for various subregions was to compare the interannual variability between the subregions. From it, we conclude that winter temperature anomalies vary at the scale of the whole Alps, while winter precipitation shows a North-South distinction, and snow depth varies differently among all the subregions.

However, we agree that the strong overlaps makes the graphs hard to read, and Taylor diagrams act as quantitative supports to the visual comparison of the interannual variation of anomalies.

According to this comment, and to several major comments above, we changed the methodology of aggregation of different elevation band (please refer to the response to comment 1.1.4 for further details), removed the shaded areas around solid lines, moved Figures 11 to 14 into appendix and replaced it by the aggregated results at the scale of the whole Alps in the manuscript.

**1.2.24 L586 : Is ERA5 really assimilating in-situ snow depth ? I thought this was not case.**

According to Hersbach et al. (2020), ERA5 assimilates in-situ snow depth observations. From this reference one reads : "In addition, the LDAS uses in situ observations of the global SYNOP network for temperature and humidity at screen level, soil moisture and snow depth. From 2004 onwards it also uses information on snow cover over the Northern Hemisphere from the multi-sensor IMS system." For further details please refer to the section 5. Observations of Hersbach et al. (2020).

**1.2.25 Unclear, how you estimate trends at the seasonal scale from monthly values. This violates the independence assumptions in the trend estimators. Why not first calculate seasonal means ?**

Indeed, we initially performed the computation based on monthly values, which violates independence assumptions. Accordingly, we now calculate the trends based on seasonal means directly, corrected it into the revised manuscript and updated Figures 15 to 18.

**1.2.26 running window trends are often hard to interpret. Maybe a time series plot might be more informative (similar to Figs 11-14 but alps-wide average) and easier to read.**

We agree that the running window trends figures are not trivial to read, but they synthesize a wide range of results concerning the evolution of climate and its internal variability that we believed highly valuable for the purpose of this section.

**1.2.27   L653 : So the differences are caused by the different LSMs and snow schemes ?**

There are three main differences (if we let asides the increased resolution of ERA5-Land) between ERA5, ERA5-Crocus and ERA5-Land that could explains the differences : the absence of coupling and surface assimilation in ERA5-Crocus and ERA5-Land, and a more detailed snow schemes in ERA5-Crocus along with a different LSM. These are the main factors from which, unfortunately, we can not apportion the different contribution in this study.

**1.2.28   Fig 17 : what are the two different N numbers ?**

We thanks reviewer 1 for his/her remark, this is an oversight that have been corrected by adding in the caption of Fig 17 : " The N-number represents the number of trends detected as significant out of the total number of trends calculated."

**1.2.29   L760ff : Wouldnt the identified cold bias also delay melt ?**

Indeed, the identified cold bias should also delay melt, but the opposite phenomenon is observed, and may be compensated by other errors from the energy balance of the snow schemes that strongly overestimates melt at the end of the season. To mention it more clearly we rephrased L.762-764 : "While the negative temperature bias should favor late melting, the opposite phenomenon is observed, which point towards other problems with the snow schemes and would require additional sensitivity studies to precisely identify the causes.".

**1.2.30   L824ff : Quite harsh a statement for most of the remote sensing people, though most climatologists would agree. Please consider rephrasing more diplomatically, since the relevance of short-term trend assessments is discussable.**

Accordingly we rephrased L.824 : "As a consequence, key data records, such as satellite observations, which only cover such a limited time span, should be used with special care when they are used for trend analysis, because this could lead to erroneous results."

**1.2.31   L853ff : Im not sure the deficiency is always on the reanalyses and models. Observational data, particularly in the Alps, has its own deficiencies. EOBS, for instance, has a very low station density south of the Alps affecting the temperature signal. In addition, observational evidence on EDW in the Alps is sparse and contradicting (some studies also show negative EDW).**

L.853 is part of a paragraph that highlights some undeniable deficiencies (due to their physical implausability) from some reanalyses concerning the evolution of seasonal indicators over the last decades. We are aware that observational datasets are not exempt from weaknesses, nonetheless, in all cases (E-OBS for temperature, in situ obs. for snow conditions and LAPrec for precipitation) they provides rather robust estimates of the evolution of these indicators over the last decades, at least at a large scale, with values that are consistent with the literature based on in situ observation.

**1.2.32   L881ff : Since you did not test these factors explicitly, you might want to consider rephrasing as potential causes that you identified.**

Accordingly, we rephased part of the paragraph L.881ff : "Among the various plausible factors, we suspect... ".

**1.2.33   Note that the HISTALP summary regionalization you used is mostly based on temperature, and especially the precip zones (and thus also snow) can differ quite strongly : ZAMG - HISTALP. This could be discussed when talking about regional results (especially precip and snow)**

Indeed, the choice of this spatial clustering is somehow arbitrary, we could also have chosen the one performed by Matiu et al. (2021), or performed our own. In many different aspects studied in our results, the results vary only slightly between the subregions, therefore most of the results are presented for the Alps as a whole and why we have choosen not to discuss this aspect.

**Références**

Bandhauer M, Isotta F, Lakatos M, Lussana C, Båserud L, Izsák B, Szentes O, Tveito OE, Frei C (2022) Evaluation of daily precipitation analyses in e-obs (v19. 0e) and era5 by comparison to regional high-resolution datasets in european regions. International Journal of Climatology 42(2) :727–747, DOI 10.1002/joc.7269

Hersbach H, Bell B, Berrisford P, Hirahara S, Horányi A, Muñoz-Sabater J, Nicolas J, Peubey C, Radu R, Schepers D, et al. (2020) The era5 global reanalysis. Quarterly Journal of the Royal Meteorological Society 146(730) :1999–2049, DOI 10.1002/qj.3803

Isotta FA, Vogel R, Frei C (2015) Evaluation of european regional reanalyses and downscalings for precipitation in the alpine region. Meteorologische Zeitschrift 24 :15–37, DOI 10.1127/metz/2014/0584

Matiu M, Crespi A, Bertoldi G, Carmagnola CM, Marty C, Morin S, Schöner W, Cat Berro D, Chiogna G, De Gregorio L, et al. (2021) Observed snow depth trends in the european alps : 1971 to 2019. The Cryosphere 15(3) :1343–1382, DOI 10.5194/tc-15-1343-2021

Monteiro D, Caillaud C, Samacoïts R, Lafaysse M, Morin S (2022) Potential and limitations of convection-permitting cnrm-arome climate modelling in the french alps. International Journal of Climatology DOI 10.1002/joc.7637

Ribes A, Corre L, Gibelin AL, Dubuisson B (2016) Issues in estimating observed change at the local scale– a case study : the recent warming over france. International Journal of Climatology 36(11) :3794–3806, DOI 10.1002/joc.4593

Sen PK (1968) Estimates of the regression coefficient based on kendall's tau. Journal of the American statistical association 63(324) :1379–1389, DOI 10.1080/01621459.1968.10480934

---

## Author Comment (AC2)

**Responses to RC2 on manuscript EGUSPHERE-2023-166**

***Authors :***
Diego MONTEIRO, Samuel MORIN

**1   Reviewer comment 2**

The manuscript compares temperature, precipitation, and snow values of the European Alps from several reanalysis and climate modeling products with remote sensing and in-situ data of the last few decades. The authors focused their extensive analysis on 4 large sub-regions and on three elevation band between 600 and 2400 m. The results show that the agreement varies widely and depends mainly on the data set or variable considered. Such analyses are an important base for many scientific investigations since more and more applications rely on such spatially gridded long-term datasets. Moreover, the study provides a very valuable overview on the strength and weaknesses of the currently available observation datasets and gridded modelling products.

The structure of the paper is clear, the text mostly concise and follows a obvious thread. The methods and results are nicely presented. Despite the many acronyms the study is relatively easy to read. Except for one major point, there are only minor issue listed below. Therefore, I suggest accepting the manuscript as soon as the following points, have been addressed :

**

We thank Reviewer 2 for his/her useful comments that have lead to an improved revised manuscript.

**1.1   Major comment**

**1.1.1   The determination of SCD, SOD and SMOD is fully dependent on complete time series. According to 2.3.1 you still allowed 20% of less of missing data in the in-situ snow depth data set. How many of the used time series were affected by such gaps and how were the above snow cover related variables determined in such cases ? What was the procedure, if two or more periods of the same length were detected ?**

As mentioned in the manuscript section 2.3.1, most of the missing values are temporally located during summer, where no snow is present. During this period, for some stations neither manual measurements nor a filling of the time series were performed. We did not check manually the temporal location of each missing values for each of the time series, but it is highly unlikely, given that we used quality-checked and gapfilled products from Matiu et al. (2021), that missing values are found during the snow season.

In order to address this point, we provide in the following a short sensitivity study to investigate the impact of using a more restrictive threshold.

For this, we removed stations that have more than 5% missing values for the entire year, allowing us to identify at which elevations the stations with 5% to 20% missing values are located. In total 74 stations were identified. Out of them, only 3 were located above 1800m.

To ensure that these series, despite their percentage of missing values (lying between 5% to 20%) provide realistic values of snow cover duration, we compare the distribution of the mean snow cover duration over the 2000-2015 period for stations that have less than 20% missing values with stations that have between 5% and 20% missing values, which we show on Figure 1. Figure 1ab show the impacts of using different thresholds on the snow cover duration distribution. We see that the blue points on Figure 1a, showing between 5 to 20% missing values, are mostly in the quantile range 25-75. Figure 1b shows the distribution of the SCD for two thresholds of missing values : 5% and 20%. As no significant differences can be observed when setting a more restrictive threshold values for the missing values, we choose to keep the initial threshold value of 20% missing values to maximize the number of observations included in the analysis.

[Figure]

FIGURE 1 – a - Boxplot representing the distribution of the mean values of snow cover duration (SCD) for the 2000-2015 period using a threshold set at 20% missing values for three elevation bands (600 m ±150 m, 1500 m ±150 m, 2400 m ±150 m). Blue points are the mean values of SCD for the 2000-2015 period for stations that have 5% to 20% missing values. b - Blue boxplots represent the distribution of the mean values of SCD for the 2000-2015 period using a threshold set at 20% missing values for three elevation bands (600 m ±150 m, 1500 m ±150 m, 2400 m ±150 m) and orange boxplots for a threshold set at 5%.

Concerning the last part of the question, for the computation of the snow cover duration, missing values are considered as a 0 (absence of snow), and we are looking for the longest consecutive snowy period. If two or more period of the same length are detected, the first is chosen to determine the snow onset date (SOD) and the snow melt out date (SMOD).

**1.2 Minor comments**

**1.2.1 L11 : Please add the information that the presented results are mainly valid for the November to April period.**

Accodingly, we added L.9 : "..., mainly for the winter period (from November through April).".

**1.2.2 L20 : Since such a sentence does not really provide any meaningful information, certainly not in the summary, delete it or be more specific.**

We find valuable to introduce the main results of the section dedicated to the interannual variability and trends comparisons between the reference and the evaluated datasets, which mention that most of the datasets provides result consistent with past literature, which is the main "summarized" conclusion of this section.

**1.2.3 L36 : Please provide reference for 5 days per decade**

The reference for "5 days per decade" is (Hock et al., 2019), that we cited above and below.

**1.2.4 L37 : Please provide an example for altering the magnitude of natural hazard in the winter season.**

From the IPCC SROCC (Hock et al., 2019), we can mention snow avalanches, floods and landslides. Accordingly we added L.37 : "... such as snow avalanches, floods and landslides".

**1.2.5 L94 : contrasting results twice ?**

The manuscript was corrected accordingly.

**1.2.6 L105 : Missing clause !**

The manuscript was corrected L. 105 : "[...] and find that the quality of the dataset for a given area is for a large part determined by the number of assimilated observations."

**1.2.7 L106 : I do not understand this last sentence. Please rephrase.**

It was due to a typing error, the manuscript was corrected accordingly L.105 : "... find that the quality of the dataset for a given area is for a large part determined by the number of assimilated observations.".

**1.2.8 L159 : Please shortly mention, why only the 2000-2015 period was considered.**

Indeed, in this section, the choice of the time period for each of the subsets was not explained. Accordingly, we modified and rephrased part of the paragraph L.151-161 : "In order to have the largest spatial coverage to compute monthly to seasonal mean snow depth over large regions, we keep all stations data that have at least 70% valid daily values over the November to April for the 1985-2015 period (see Figure 1b). Section 2 focuses on the snow cover seasonality, using the indicators SCD, SOD and SMOD computed using continuous daily values of snow depth over the winter period, and compare it to satellite observations from MODIS (record starting in 2000). Most of the missing values happen in summer (for most of the observation stations, no snow is present during this period). In this section, we keep stations with more than 80% of valid daily values over all year of the 2000-2015 period (see Figure 1c). Section 3 focuses on the interannual variability and trends. This requires the most homogeneous possible dataset along with a sufficiently long time period, so we only keep stations with more than 90% valid daily values over the winter months (November to April) of the 1968-2018 period (see Figure 1d)."

**1.2.9 Fig.2 : Yellow color is hard to read. Id suggest adding (e.g. right of the color box) the final spatial and temporal resolution of each product.**

Yellow color has been changed to a darker yellow, easier to read. Spatial and temporal resolution are already indicated within the squares representing each of the component of the products.

**1.2.10 L302 : Please rephrase**

Accordingly, part of the paragraph L.300-305 has been rephrased : "The agreement metrics used are skill scores based on the confusion matrices calculated using daily values of presence or absence of snow, considering in situ observations as the truth :

— True Positive Rate (TPR) corresponding to the proportion of number of points flagged as presence of snow in MODIS pixel and in the corresponding in situ station.
— True Negative Rate (TNR) corresponding to the proportion of number of points flagged as absence of snow in MODIS pixel and in the corresponding in situ station.

— Positive Predictive Value (PPV) or Precision is the proportion of number of points correctly flagged as presence of snow in MODIS pixel.
— Negative Predictive Value (NPV) is the proportion of number of points correctly flagged as absence of snow in MODIS pixel.
— Accuracy corresponding to the proportion of the total number of predictions that were correct (both presence and absence of snow).

"

Then, differences for the different thresholds between MODIS and stations observations for our three indicators (SCD, SOD, MOD) are quantified using mean absolute error (MAE) and mean error (ME) values."

**1.2.11   L217 : You mention several NH snow analyses, but reference only one, which is about arctic snow only ?**

We added an other reference (Decharme et al., 2016), that investigates the performance of the surface reanalysis over Eurasia.

**1.2.12   L275 : Please shortly mention what the reference is.**

Accordingly, we modified L.275 : "... compared to the DEM at 100 m".

**1.2.13   L287 : Last section of what ?**

"Last section" was not precise, we modified it L.287 : "Section 3.3".

**1.2.14   Fig.3 : What is UE Alps ?**

UE Alps incorrectly stood for European Alps, but it was unclear, so we replaced it by : "Whole Alps".

**1.2.15   L355 :..there are not much differences in the results at the sub-regional scale. This contradicts the results of Matiu et al. (2021). Please explain.**

When saying that there are not much differences in our results at the subregional scale, we refers to section 3.1 only, that adressed the differences between the evaluated datasets and the reference one concerning the mean characteristics of the snow cover. Matiu et al. (2021) show differences between subregions of the mean characteristics of the snow cover for its in situ observational dataset, which differs what we investigated in our study.

**1.2.16   L369 : normalized mean error. In the same sentence you also use relative mean error ?**

In the study, both terms refer to the use of normalized (by mean values) error metrics and expressed in pourcentage. Using both terms can be confusing, so we only keep the term "normalized" for the error metrics.

**1.2.17   Fig.5 : I do not see circle markers ? Figure caption : Sub figures a,b,c labeled wrong order.**

Circle markers are only associated with the black line (representing the observations), and are rather small to let visible other lines. The labeling was designed to be ordered from the top to the bottom and therefore is not in the wrong order. Nonetheless, according to the comment 2.2.24, we reversed this order in the revised manuscript.

**1.2.18   Fig.6 : Please add units to the different column heads. Figure caption : What about the unit of the scores of temperature ?**

Accordingly we added units to the column heads.

**1.2.19   Fig.8 : Labels of x- and y-axis of right side figures are too small.**

Accordingly, labels of Figure 8 were enlarged.

**1.2.20   L453 : Please rephrase**

Accordingly we rephrased L.453 : "Overall, most of the datasets provide winter precipitation rather close to the reference, excepted climate simulations CNRM-AROME and CNRM-ALADIN that strongly overestimate it at intermediate and high elevations."

**1.2.21   L457 : could be nuanced ? Please rephrase**

Accordingly, we rephrased L.457 : "However, the origin of the overestimation also identified for the other datasets during the winter period may be for a part due to LAPrec deficiencies."

**1.2.22   L495 : consistent with boxplot ? Only if we assume that the negative whisker is caused by the low elevation pixels. Please explain.**

The sentence was lacking clarity in its initial form, so we rephrased L494-495 : "For CNRM-AROME, the map on Figure **??**a seems to show an elevational pattern with a slight underestimation of the snow cover duration in valleys and an overestimation elsewhere. The boxplot Figure **??**b confirms it, revealing centered around 0 biases at low elevation, and a generalized overestimation above.".

**1.2.23   Fig.9 : Caption : In order to compared.. ?**

We rephrased part of the sentence in caption : "Note that MODIS products initially at 500 m horizontal resolution have been reggrided over each dataset grid using a first-order conservative method.". We decided to leave this sentence as a reminder of the methodology used to construct it.

**1.2.24   L503 : ..underestimated in all datasets.. Where to see ? Fig.10 : The order of a,b,c (a for the lowest plot) is different from all the other figures, but I like it more.**

According to this comment and the comment 1.1.4 from reviewer 1, we rephrased part of the text L.503 : "From Figure 10abc, the size of the errorbars provides information on the spatial variability of the SOD and the SMOD. Based on this information, we can conclude that it is underestimated in all datasets, except CNRM-AROME, but this may be partly related to a too small horizontal resolution in most of the datasets."

**1.2.25   L537 : Solid line with circle markers should be part of the figure caption. What is the geographical mean value ?**

Accordingly this part has been removed from the text. The "geographical mean value" stands for the mean value over the subregion. This expression has been modified for better clarity.

**1.2.26   Fig.11 : What is the reason to show to markers ? The different markers can not be separated from each other. For comparability Id suggest to use the same axis scale for all four Taylor diagrams.**

In our opinion, markers help to differenciate the different lines and identify the precise value associated with a specific date.

**1.2.27   Fig. 12 : What is the reason for the large difference between the red and blue line for the second last year ?**

It was an artifact that has been corrected in the revised manuscript.

**1.2.28   L572 : lower**

The manuscript have been corrected accordingly.

**1.2.29   L582 : anomaly values**

The manuscript have been corrected accordingly.

**1.2.30   L599 : ..precipitation for the 1950-2020 period**

We can not add it to the sentence, only the ERA5 dataset spans the 1950-2020 period, it would therefore be inaccurate.

**1.2.31   L640 :  over the last decade only ?**

It was a typing error leading to the missing of the plural, we corrected it : "... over the last decades.".

**1.2.32   Fig.15 : Winter (Nov-Apr) trend values**

The manuscript has been corrected accordingly.

**1.2.33   Fig.17 : What is the meaning of the two N-values for each dataset ? N-values are too small !**

We thanks reviewer 2 for his/her remark, this is an oversight that has been corrected by adding in the caption of Fig 17 : " The N-number represents the number of trends detected as significant out of the total number of trends calculated.".

**1.2.34   Fig.17 : Caption :  winter (Nov-Apr) trend grid points included for three elevation bands (+/- 150 m) given on the y-axis.**

We thanks reviewer 2 for his/her remark and corrected the caption accordingly.

**Références**

Decharme B, Brun E, Boone A, Delire C, LeăMoigne P, Morin S (2016) Impacts of snow and organic soils parameterization on northern Eurasian soil temperature profiles simulated by the ISBA land surface model. The Cryosphere 10(2) :853–877, DOI 10.5194/tc-10-853-2016

Hock R, Rasul R, Adler C, Cáceres B, Gruber S, Hirabayashi Y, Jackson M, Kääb A, Kang S, Kutuzov S, Milner A, Molau U, Morin S, Orlove B, Steltzer H (2019) High Mountain Areas. In : Pörtner HO, Roberts D, Masson-Delmotte V, Zhai P, Tignor M, Poloczanska E, Mintenbeck K, Alegriáa A, Nicolai M, Okem A, Petzold J, Rama B, Weyer N (eds) IPCC Special Report on the Ocean and Cryosphere in a Changing Climate, ., pp 131–202

Matiu M, Crespi A, Bertoldi G, Carmagnola CM, Marty C, Morin S, Schöner W, Cat Berro D, Chiogna G, De Gregorio L, et al. (2021) Observed snow depth trends in the european alps : 1971 to 2019. The Cryosphere 15(3) :1343–1382, DOI 10.5194/tc-15-1343-2021

---

## Referee Report (RR1)

**Review of Monteiro and Morin:**

**Multi-decadal evaluation and analysis of past winter temperature, precipitation and snow cover information over the European Alps from reanalyses and climate models**

**General comment:**

This is a very relevant study, which compares in a particularly comprehensive way data sets of winter temperature, winter precipitation, and snow depth (or the derived variable of snow cover duration) for the Alpine region. The data sets are on the one hand products derived from station data, satellite products, reanalysis data and climate simulations. Such a comprehensive and in-depth comparison is, to my knowledge, unique and of great value to the cryospheric research community but also well beyond. Since the Alps are characterized by a high density of measurements, satellite products, modeling studies, and investigations of spatiotemporal changes, the results can be considered as a benchmark of comparison between observational data and model results for mountain regions.

**Minor comments:**

I have only several minor comments or suggestions for improvement:

Title: The title could be more concise. Actually, it is more a comparison of the data sets (as it is also written in the abstract) than an evaluation. I suggest also to make clear in the title that all climate variables studied have as reference the winter period (so also for precipitation). Also, instead of "snow cover information", one could use "snow depth information" to make clear that the study refers to the quantity "snow depth" as well as quantities derived from it.

Proposal for the title: "Comparison and analysis of winter temperature, winter precipitation and snow depth variables in the European Alps from multiple datasets".

**Abstract:**

The data on trends in snow cover duration given in the abstract are altitude dependent (as will be shown later in the paper), so giving a range of variation rather than a fixed value would be more accurate and clearer to the reader.

From the sentence "Reference datasets and some of the evaluated datasets provides past trends in line with current available literature" it remains unclear what is meant. Which datasets were evaluated? (In this study or elsewhere).

I find the altitude dependencies of the trends (elevation dependent climate change) for temperature, precipitation and snow sizes a relevant result and suggest to include it in the abstract as well.

**Objective of the study as described in the introduction section:**

It is stated (page 4/118-120): *The objective of the present study is to compare the performance of different datasets from different modelling strategies in the European Alps, in order to provide the best possible estimate of the state of the snow cover, and its first order drivers, wintertime near surface temperature and precipitation.*

I recommend to rethink if this is really the main objective:

- best possible estimate of state of snow cover (and its first order drivers)
- comparison of the performance of different datasets from different modelling strategies

If the objective is really for best possible estimate then this aim should be more reflected in the conclusions. Moreover, comparison includes not only the modelling data. I guess, this is certainly already a special fine-tuning of the study, but it would strengthen the work once again.

**Data and methods:**

2.4.4 Time periods, statistics and trend analyses

Please clarify in the formula for correlation what the "n" means.

**Conclusions**

Another suggestion from my side concerns the Conclusions. My impression about this is that the paper does very nicely the comparison between the different datasets and also shows the spatiotemporal trends of the datasets for temperature, precipitation and different snow variables, but the conclusions from this are still somewhat open. I totally agree with the statement that none of the datasets outperforms the others. But ev. it could be interesting to contrast the detected differences between the data sets with the detected trends. This could be done either in a figure (however there is already an extremely high supply of figures and should not be enlarged) or purely textual, where the latter is probably easier to do.

**Figures:**

Figure 1 caption:  … at 1km and contour of the Alpine Convention outline of the Alps and the four ….

For Figures 7, 8, 9 14, A1, A2, A3 it would be helpful for the reader if the structure of the three altitude bands would be more emphasized in a graphic way. For some of the Figures, one could get the impression that the Y-axis is a continuous representation of the altitude (but defacto it is only an indication of the altitude bands).

To some of the Figures the font size is already very small and it is not easy to read (but ev. it still meets the Copernicus requirements)

I recommend to improve the language quality by looking for repetition of words and some spelling errors as well as simplifying sentence order / improving readability.

Example e.g. page 4 last paragraph : We investigate … We take …. We also exploit …. By doing so, we aim …

---

## Author Response (AR2)

**Responses to RC3 on manuscript EGUSPHERE-2023-166**

*Authors :*
Diego MONTEIRO, Samuel MORIN

**1 Reviewer comments 3**

This is a very relevant study, which compares in a particularly comprehensive way data sets of winter temperature, winter precipitation, and snow depth (or the derived variable of snow cover duration) for the Alpine region. The data sets are on the one hand products derived from station data, satellite products, reanalysis data and climate simulations. Such a comprehensive and in-depth comparison is, to my knowledge, unique and of great value to the cryospheric research community but also well beyond. Since the Alps are characterized by a high density of measurements, satellite products, modeling studies, and investigations of spatiotemporal changes, the results can be considered as a benchmark of comparison between observational data and model results for mountain regions.

**

We thank reviewer 3 for his/her comprehensive evaluation and positive appreciation of our work.

**1.1 Title**

I have only several minor comments or suggestions for improvement :

**1.1.1 The title could be more concise. Actually, it is more a comparison of the data sets (as it is also written in the abstract) than an evaluation. I suggest also to make clear in the title that all climate variables studied have as reference the winter period (so also for precipitation). Also, instead of "snow cover information", one could use "snow depth information" to make clear that the study refers to the quantity "snow depth" as well as quantities derived from it.**
**Proposal for the title : "Comparison and analysis of winter temperature, winter precipitation and snow depth variables in the European Alps from multiple datasets".**

We thank reviewer 3 for his/her suggestion regarding the title. We agree that it could be more concise, but we should not remove some informative words. Indeed, the proposal does not mention the temporality of the period studied "multi-decadal" and "past", which we find very useful and have decided to keep in the title.

The word "evaluation" refers to a comparison between simulated fields and reference fields. Most of our analyses in the article are comparisons where we take the observational datasets as references, even though we are aware of their own limitations and mention them. Nevertheless, it's not clear whether or not this word adds valuable information to the title, so we've decided to delete it.

Concerning the use of "snow cover information" instead of "snow depth information", we have decided to keep it, as it may be clearer to readers that snow depth is not the only snow variable being analyzed, but also the snow cover onset and melt-out dates and snow cover duration.

Accordingly, we propose a new title for the article, as a compromise between clarity, informativeness and size. Note that the title of the original manuscript was shorter, and has been extended following suggestions from reviewer 1.

"Multi-decadal analysis of past winter temperatures, precipitation and snow cover data in the European Alps from reanalyses, climate models and observational datasets".

**1.2 Abstract**

**1.2.1 The data on trends in snow cover duration given in the abstract are altitude dependent (as will be shown later in the paper), so giving a range of variation rather than a fixed value would be more accurate and clearer to the reader.**

We thank reviewer 3 for his/her remark, and rephrased part of the paragraph L.22 to 25 in order to include information about the elevational gradient of the trend :

"Based on these datasets, over the last 50 years (1968-2017) at a regional scale, the European Alps have experienced a winter warming of 0.3řC to 0.4řC per decade, stronger at lower elevation and a small reduction of winter precipitation, homogeneous with elevation. The decline of the winter snow depth and snow cover duration range from -7% to -15% per decade and from -5 days to -7 days per decade, respectively, both showing a larger decrease at low and intermediate elevation."

**1.2.2   From the sentence "Reference datasets and some of the evaluated datasets provides past trends in line with current available literature" it remains unclear what is meant. Which datasets were evaluated ? (In this study or elsewhere).**

Accordingly, we rephrased L21 of the abstract for clarity : "Nevertheless, many of the considered datasets in this study exhibit past trends in line with the current state of knowledge."

**1.2.3   I find the altitude dependencies of the trends (elevation dependent climate change) for temperature, precipitation and snow sizes a relevant result and suggest to include it in the abstract as well.**

We have taken this suggestion into account and suggest above a modification of the abstract.

**1.2.4   Objective of the study as described in the introduction section :**
**It is stated (page 4/118-120) : The objective of the present study is to compare the performance of different datasets from different modelling strategies in the European Alps, in order to provide the best possible estimate of the state of the snow cover, and its first order drivers, wintertime near surface temperature and precipitation.**
**I recommend to rethink if this is really the main objective :**
**- best possible estimate of state of snow cover (and its first order drivers)**
**- comparison of the performance of different datasets from different modelling strategies**
**If the objective is really for best possible estimate then this aim should be more reflected in the conclusions. Moreover, comparison includes not only the modelling data. I guess, this is certainly already a special fine-tuning of the study, but it would strengthen the work once again.**

We thank reviewer 3 for this comment. Indeed, the conclusion does not indicate specific datasets or even strategies for constructing datasets (i.e. from models and observations, using assimilation or not...) that would obtain better results than others. However, we aim to answer that some strategies have advantages when it comes to representing daily and/or seasonal values of variables, while others are more robust when it comes to reconstructing past trends and variability.

Concerning the comparison, the first objective of the study remains to characterize the robustness of each dataset evaluated in multiple aspects of the climatology, using observational or observation-based datasets as reference. Indeed, we are aware that these datasets also have their limitations, and the study can be seen as a cross-evaluation of the "evaluated" datasets and the "reference" datasets.

For clarity, we proposed to rephrase L.118-120 : "The objective of the present study is to compare the performance of different datasets from different modelling strategies in the European Alps, in order to better understand their different characteristics and assess how to provide the best possible estimate of the snow cover spatio-temporal variability and trends, and its first order drivers, wintertime near surface temperature and precipitation."

**1.3   Data and methods**

**1.3.1   2.4.4 Time periods, statistics and trend analyses Please clarify in the formula for correlation what the "n" means.**

Indeed, as the meaning of "n" is unclear, we have replaced it in the equation with "N", in reference to the sample size as indicated.

**1.4 Conclusions**

**1.4.1 Another suggestion from my side concerns the Conclusions. My impression about this is that the paper does very nicely the comparison between the different datasets and also shows the spatiotemporal trends of the datasets for temperature, precipitation and different snow variables, but the conclusions from this are still somewhat open. I totally agree with the statement that none of the datasets outperforms the others. But ev. it could be interesting to contrast the detected differences between the data sets with the detected trends. This could be done either in a figure (however there is already an extremely high supply of figures and should not be enlarged) or purely textual, where the latter is probably easier to do.**

We thank Reviewer 3 for this suggestions. Indeed, a more in-depth analysis linking the multiple aspects examined in the study for each of the datasets would be interesting. Nevertheless, we feel that at this stage, the manuscript is already well-furnished, if not too well-furnished, and we have therefore decided not to add any further analysis and leave this for further investigations.

**1.5 Figures**

**1.5.1 Figure 1 caption : ... at 1km and contour of the Alpine Convention outline of the Alps and the four .**

The captions have been corrected accordingly.

**1.5.2 For Figures 7, 8, 9 14, A1, A2, A3 it would be helpful for the reader if the structure of the three altitude bands would be more emphasized in a graphic way. For some of the Figures, one could get the impression that the Y-axis is a continuous representation of the altitude (but defacto it is only an indication of the altitude bands).**

We understand that the discontinuous y-axis containing numerous boxplots for each elevation band may be rather unusual for some readers, but we believe that the caption provides sufficiently detailed information to avoid any misunderstanding of the figure's meaning.

**1.5.3 To some of the Figures the font size is already very small and it is not easy to read (but ev. It still meets the Copernicus requirements)**

Upon acceptance of the manuscript we will interact with Copernicus, if need be, to adjust the figures where needed.

**1.5.4 I recommend to improve the language quality by looking for repetition of words and some spelling errors as well as simplifying sentence order / improving readability. Example e.g. page 4 last paragraph : We investigate ... We take .... We also exploit .... By doing so, we aim ..**

Manuscripts accepted for publication in Copernicus journals are usually edited by professionnal editors at the copy-editing stage, which will ensure than such issues are resolved